# Economic and environmental competitiveness of multiple hydrogen production pathways in China

Guangyao Fan[1], Hui Zhang [1], Bo Sun [1] ✉ & Fengwen Pan [2,3] ✉

This study utilises the optimization method to ascertain the levelized cost of hydrogen and life cycle carbon emissions of four water electrolysis hydrogen production systems across 31 provinces and regions of China, and compares these with hydrogen production from coal, natural gas and industrial by-products. The findings indicate that the grid-connected water electrolysis hydrogen production system has low-carbon advantages only in certain provinces, and time-of-use electricity prices can improve its economic competitiveness. The off-grid water electrolysis hydrogen production system can achieve near-zero carbon emissions, although additional investment is required to configure larger capacities for electricity energy storage and hydrogen storage. Projections indicate that by 2045–2050, this system could emerge as the most cost-effective to hydrogen production, a milestone that could be advanced by 5–15 years through the implementation of specific carbon reduction incentives or production subsidies. Prior to this timeframe, hydrogen production through industry by-products emerges as a viable alternative for the development of hydrogen energy.

Hydrogen is a secondary energy with abundant sources, with global production currently exceeding 95 million tons[1]. In the future hydrogen energy may achieve a wide range of applications[2,3], promoting a high share of renewable energy development and accelerating the decarbonization of industry, transportation, buildings and other sectors[4,5]. According to predictions, the demand for hydrogen energy could increase to 660 million tons by 2050, account for 22% of global terminal energy demand. At the same time, 80 billion tons of carbon dioxide could be reduced cumulatively, which is 11% of the carbon reduction needed to limit global warming to 1.5–1.8 °C[6]. Therefore, hydrogen energy is considered an important carrier for the deep decarbonization of the global energy system.

Up to the present, over 50 countries and regions around the world have included hydrogen in their energy strategies[7]. China has actively promoted the development of hydrogen energy industry to facilitate the achievement of the carbon peaking and carbon neutrality goals[8]. Mid-to-Long-Term Plan for the Development of Hydrogen Energy

Industry (2021–2035) issued in March 2022 explicitly outlined the development goals for hydrogen energy[9]. In April 2024, the Energy Law of the People's Republic of China (Draft) was promulgated, formally incorporating hydrogen energy into the realm of energy resources[10]. In addition, hydrogen energy has been integrated into the 14th Five-Year Plan across 30 provincial-level administrative regions nationwide[11,12]. Currently, China's hydrogen production has exceeded 50 million tons, making it the world's largest hydrogen producer[9]. However, 81% of the its hydrogen production comes from fossil fuel, i.e. coal and natural gas, accounting for 62% and 19% respectively[13]. It is worth noting that to achieve the goal of carbon reduction through hydrogen energy, it must be produced by water electrolysis through low-carbon electricity or renewable electricity. However, only 1% of hydrogen is currently sourced from water electrolysis.

Currently, the feasibility of water electrolysis hydrogen production has been researched extensively. The International Energy Agency (IEA)[1] compared the costs and carbon emission intensity of different

[1]School of Control Science and Engineering, Shandong University, Jinan, Shandong 250061, China. [2]Weichai Power Co., Ltd., Weifang 261061, China. [3]National Center of Technology Innovation for Fuel Cell, Weifang 26100, China. ✉e-mail: sunbo@sdu.edu.cn; panfengwen@outlook.com

hydrogen production pathways from a global perspective. The results showed that the cost of hydrogen production depends on the technology and cost of the energy source used, which typically varies considerably by region. The cost of water electrolysis hydrogen production system exceeded 3.4 $/kg $H_2$, with a carbon emission intensity of 24 kg $CO_2$/kg $H_2$ for using grid electricity, and close to zero for using renewable energy electricity without considering the manufacturing of photovoltaics or wind turbines. Guerra et al.[14] focused on 20 states in the United States, simulated the operation of electrolyzer dynamically and demonstrated that electrolysis equipment can provide hydrogen with cost competitiveness. Glenk et al.[15] pointed that the cost of hydrogen production from renewable energy in Germany is 3.23 €/kg, already achieving cost competitiveness. For India's carbon neutrality goal, Song et al.[16] indicated that water electrolysis hydrogen production can be utilized to reduce the cost of energy system by 10% in 2050 and promote the decarbonization of industry and the power sector.

With the breakthrough of renewable energy and the technology of electrolyzer in China, many scholars have begun to explore the economic and environmental feasibility of water electrolysis hydrogen production in China. The cost and carbon emissions of various hydrogen production pathways in China were compared by IEA firstly, including hydrogen production from grid electricity, renewable energy electricity, natural gas and coal[17]. Yang et al.[18] investigated the potential role of clean hydrogen by focusing on the industries facing bottlenecks in carbon reduction in China, i.e., heavy industry and heavy transportation. The results showed that clean hydrogen can substantially reduce carbon emissions from heavy industry. And about 1.72 trillion dollars in new investments can be avoided in clean hydrogen scenario that reaches 65.7 million tons of production in 2060. Song et al.[19] explored the feasibility of transporting hydrogen in liquid form to Japan, which produced by offshore wind power electrolysis in China. The results showed that China's offshore energy deliveries and costs are in line with Japan's idealized future projections. Pan et al.[20] provided a detailed assessment of the cost of hydrogen production system combined photovoltaic and grid in China, using a fixed daily supply of hydrogen as a requirement. Li et al.[21] compared the average cost of hydrogen production from electricity and coal in China using a hydrogen optimization model with monthly supply and demand balances. Qiu et al.[22] calculated the total hydrogen demand in China from 2017 to 2060 using an annual supply and demand balance energy planning model.

However, the above researches were conducted only on a national spatial scale for a particular type of water electrolysis hydrogen production system, without considering the differences in system configurations and provincial resources. Firstly, electricity for water electrolysis hydrogen production system may be derived from the grid or renewable energy sources, leading to different configurations and cost. Secondly, there are notable differences in renewable resources, energy prices, and grid facilities among provinces of China. Finally, provincial hydrogen energy plans and development policies, such as Mid-to-Long-Term Plan for Hydrogen Energy of Shandong Province[23], Implementation Plan for Development of the Hydrogen Energy Industry of Beijing (2021–2025)[24], are reviewed and approved by the China National Energy Administration based on the resource endowment, energy development capacity, and environmental carrying capacity of each province. Therefore, it is of great significance to evaluate water electrolysis hydrogen production system with different configurations in all provinces of the China in this context.

Here, a model for configuration and full-year hourly operation optimization of hydrogen production systems is developed considering the hourly hydrogen supply reliability. The data are derived from real-world electricity prices and grid carbon emission factors for 31 provinces, as well as hourly photovoltaic power generation from the National Renewable Energy Laboratory. The system investment and operation schemes in different provinces are jointly optimized under

different subsidy policies for different target years. Then, the levelized cost of hydrogen (LCOH) and life cycle carbon emissions (LCCE) of different water electrolysis hydrogen production systems are quantified for 31 provinces in China, including grid, photovoltaic and grid combined, fixed renewable energy penetration rate, and off-grid. The development of multiple hydrogen production pathways in the future scenario from 2025 to 2050 is discussed. The changes of the development pathways under three actual subsidy policies, i.e., electricity price concession, investment matching incentives for carbon reduction, production subsidies, are analyzed.

In this work, the analysis shows that the current LCOH of four water electrolysis hydrogen production systems in the provinces of China are 4.6 $/kg $H_2$-7.9 $/kg $H_2$, 4.6 $/kg $H_2$-7.6 $/kg $H_2$, 5.9 $/kg $H_2$-10.1 $/kg $H_2$, and 7.3 $/kg $H_2$-14.8 $/kg $H_2$, respectively. The LCOH is generally higher in the eastern and southern coastal areas, and is relatively lower in the western and northern regions. The implementation of time-of-use electricity prices can reduce the LCOH of the grid-connected water electrolysis hydrogen production system by 0.18 $/kg $H_2$-0.90 $/kg $H_2$. The LCCE of the four systems are 5.2 kg $CO_2$/kg $H_2$-59.3 kg $CO_2$/kg $H_2$, 4.3 kg $CO_2$/kg $H_2$-47.4 kg $CO_2$/kg $H_2$, 3.7 kg $CO_2$/kg $H_2$-30.8 kg $CO_2$/kg $H_2$, and 2.27 kg $CO_2$/kg $H_2$-2.37 kg $CO_2$/kg $H_2$, respectively. Grid-connected water electrolysis hydrogen production systems have higher LCCE in the north, and have low carbon advantages only in some provinces, such as Qinghai, Tibet, Sichuan, and Yunnan. A total of 46.1 billion dollars of investment is required for China to achieve a hydrogen production structure transformation to 15% renewable hydrogen, which cumulatively reduces carbon emissions by 61.9 Mt. The LCOH of off-grid water electrolysis hydrogen production system is expected to be reduced to 2.2 $/kg $H_2$ by 2045–2050, making it the most economical pathway to produce hydrogen, and certain carbon reduction incentives or production subsidy may enable this to happen 5–15 years earlier. In summary, the optimization results can provide guidance for investors in hydrogen production projects, the evaluation results of hydrogen production system in different provinces can provide reference for national and provincial energy policy makers.

Here, we show two important insights on the development of the global hydrogen production industry: (1) Grid-connected water electrolysis hydrogen production has economic advantages only in areas where renewable energy is abundant or electricity prices are low, where electricity price concessions and production subsidy policies can reduce LCOH. (2) Off-grid water electrolysis hydrogen production system may be the most economical and low-carbon pathway in the future, with carbon reduction incentives and production subsidy policies accelerating the realization. Before it becomes economically advantageous, hydrogen production through industrial by-products is a good alternative.

## Results
### Hydrogen production pathways and optimization models
The current hydrogen production pathways in China are comprehensively reviewed, including the hydrogen production from water electrolysis, coal, natural gas, and industry by-product. Four water electrolysis hydrogen production systems are considered, as shown in Fig. 1.

System 1 (WE1): the grid hydrogen production system.

System 2 (WE2): the hydrogen production system combined photovoltaic and grid.

System 3 (WE3): the hydrogen production system with fixed renewable energy penetration.

System 4 (WE4): the off-grid hydrogen production system.

The specific description of the system can be found in Methods.

In addition, the coal gasification (CG) and the CG combined with carbon capture, utilization and storage (CG + CCUS) are selected to represent hydrogen production from coal, steam methane reformer (SMR) and SMR combined with carbon capture, utilization and storage

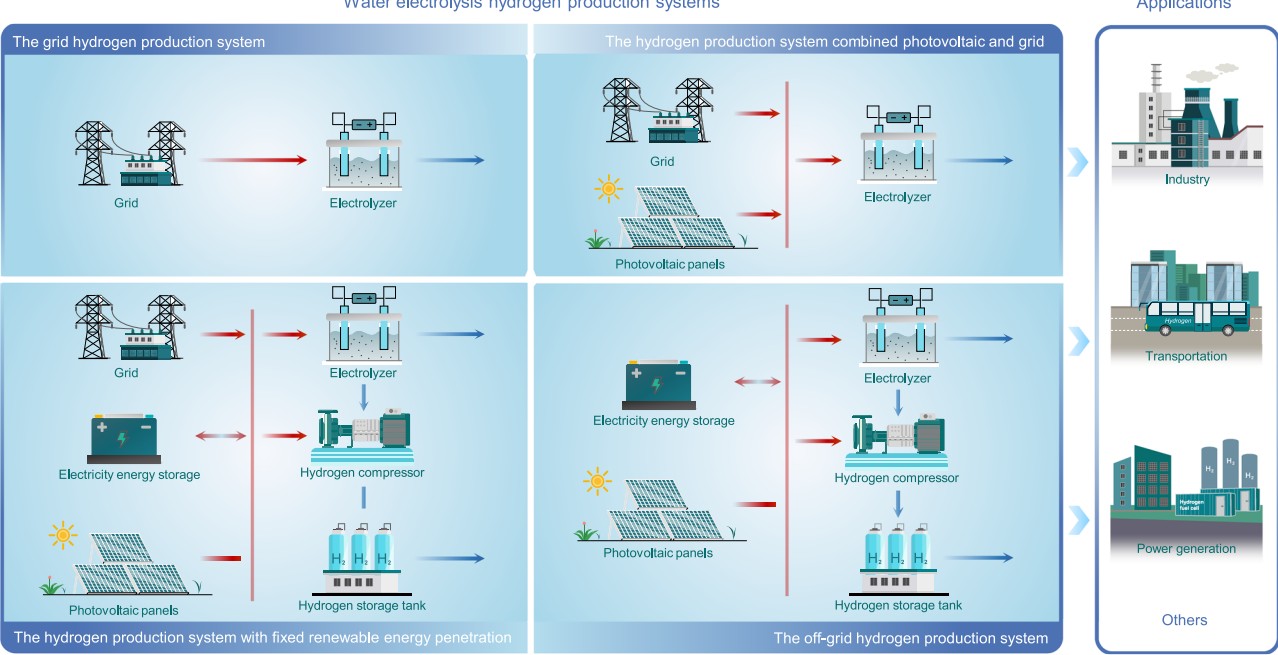

**Fig. 1 | Water electrolysis hydrogen production systems.** The red and blue lines represent electricity and hydrogen flow, respectively.

(SMR + CCUS) are selected to represent hydrogen production from natural gas. Descriptions and assumptions related to hydrogen production from coal, natural gas, and industry by-products are provided in Methods. The optimization model is constructed to optimize the configuration and operation of each system with the objective of minimizing the LCOH. To assess the environmental performance of the hydrogen production pathways, the LCCE are calculated. More details can be found in Methods.

## Economic and environmental competitiveness status
The LCOH and LCCE of multiple hydrogen production pathways in 31 provinces of China is shown in Fig. 2. From the perspective of different provinces, there are notable differences in the LCOH and LCCE of water electrolysis hydrogen production system across 31 provinces. The LCOH of the WE1 is 4.6 \$/kg $H_2$-7.9 \$/kg $H_2$, and the LCOH of the WE2 is 4.6 \$/kg $H_2$-7.6 \$/kg $H_2$. The lower and upper limits correspond to Qinghai and Shanghai respectively. It is worth noting that the economics of the WE2 is better than that of the WE1 in 18 provinces or regions including Beijing, Tianjin, Shanghai, Liaoning, Jilin, Heilongjiang, Inner Mongolia, Shanxi, Hebei, Shandong, Tibet, Guangxi, Guangdong, Zhejiang, Anhui, Jiangsu, Henan, and Hainan. In the remaining 13 provinces and regions, there are more economic advantages using the grid to produce hydrogen individually. The LCCE of water electrolysis hydrogen production system vary more substantially across provinces relative to the LCOH. The LCCE of the WE1 are 5.2 kg $CO_2$/kg $H_2$-59.3 kg $CO_2$/kg $H_2$, with the lowest in Tibet and the highest in Hebei. The LCCE of the WE2 are 4.3 kg $CO_2$/kg $H_2$-47.4 kg $CO_2$/kg $H_2$, with the lowest and highest in Tibet and Ningxia respectively. In the 18 provinces or regions above that are suitable for the configuration of photovoltaic panels, the LCCE of WE2 are reduced relative to WE1, e.g. the LCCE in Hebei are reduced by 15.1 kg $CO_2$/kg $H_2$. In addition, SMR has the lowest LCOH in Inner Mongolia at 1.3 \$/kg $H_2$, and the highest LCOH in Henan at 3.6 \$/kg $H_2$, which is determined by the natural gas price. The LCOH of SMR + CCUS is slightly higher than that of the SMR, and the LCCE is lower than that of the SMR.

From the perspective of different hydrogen production pathways, the LCOH of water electrolysis hydrogen production in each province is higher than that of hydrogen production from coal, natural gas, and industry by-product. The most economical pathway to produce hydrogen for Inner Mongolia, Qinghai, and Xinjiang is SMR, while the cost of industry by-product is the lowest in the rest of the provinces or regions. The LCCE of water electrolysis hydrogen production in most provinces are higher than those of CG. What is striking is that the LCCE of water electrolysis hydrogen production are lower than CG in seven provinces or regions including Qinghai, Tibet, Sichuan, Yunnan, Guangdong, Hubei, and Hainan. Therefore, water electrolysis hydrogen production can be given priority for development in these provinces. Among them, the LCCE of the WE2 in Guangdong and Hainan are lower than those of the CG. However, the LCCE of WE1 in these two provinces are higher than those of the CG. Therefore, the two provinces should develop water electrolysis hydrogen production through the system combined photovoltaic and grid. In addition, the LCCE of water electrolysis hydrogen production in Qinghai, Tibet, Sichuan, and Yunnan are lower than those of CG + CCUS and industry by-product. This is because that the proportion of renewable energy of the grid in these provinces is higher, and carbon emission factors for electricity purchase is lower, which are 0.095 kg $CO_2$/kWh, 0.095 kg $CO_2$/kWh, 0.117 kg $CO_2$/kWh, 0.446 kg $CO_2$/kWh. Therefore, the four provinces or regions have the most obvious advantages in terms of developing water electrolysis hydrogen production in China.

## Spatial distribution of costs and carbon emissions
The spatial distribution of LCOH for four water electrolysis hydrogen production systems is shown in Fig. 3. The LCOH of four water electrolysis hydrogen production systems is generally high in the eastern and southern coastal areas, and relatively low in the western and northern regions. The LCOH of WE1, WE2, and WE3 are determined by the electricity price and photovoltaic power generation. The LCOH of WE4 is completely determined by photovoltaic power generation, so the distribution of LCOH is opposite to the distribution of photovoltaic power generation, which means that provinces or regions with large photovoltaic power generation have low LCOH. The proportions of electricity purchase costs, investment costs, operation and maintenance, and carbon emission costs in WE1 and WE2 decrease in sequence. The electricity purchase cost accounts for 71.1%–82.3% of the total hydrogen production cost in WE1, and 58.0%–81.0% in WE2.

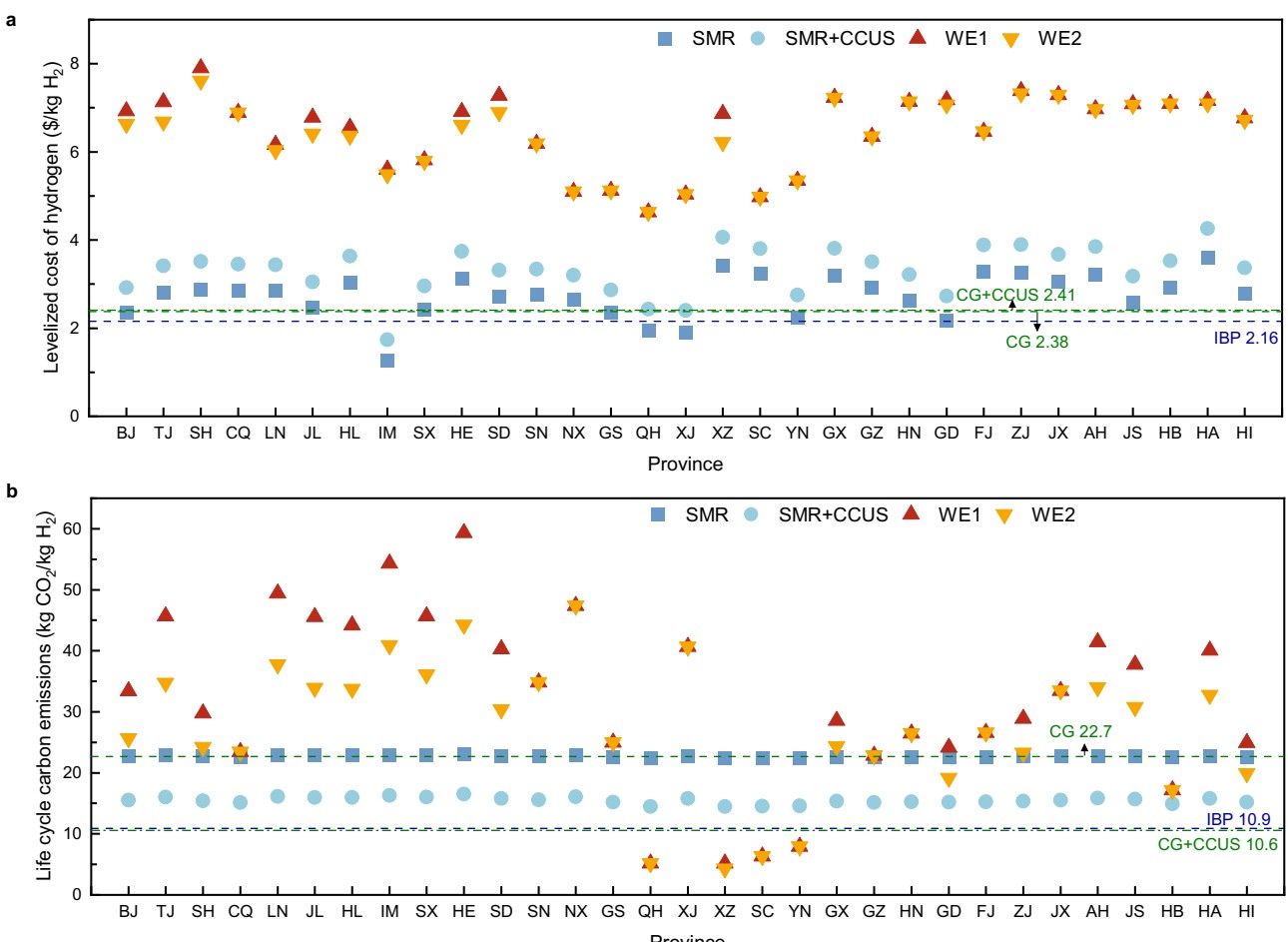

**Fig. 2 | The levelized cost of hydrogen and life cycle carbon emissions of multiple hydrogen production pathways in 31 provinces of China. a, b** Panels are the levelized cost of hydrogen and life cycle carbon emissions, respectively. WE1 and WE2 as examples of the water electrolysis hydrogen production system, WE1 refers to the grid hydrogen production system, WE2 refers to the hydrogen production system combined photovoltaic and grid. SMR refers to steam methane reforming. SMR + CCUS refers to steam methane reforming combined with carbon capture, utilization and storage. CG refers to coal gasification. CG + CCUS refers to coal gasification combined with carbon capture, utilization and storage. IBP refers to the hydrogen production from industrial by-products. The abbreviations of provinces are listed as follows: Beijing (BJ), Tianjin (TJ), Shanghai (SH), Chongqing (CQ), Liaoning (LN), Jilin (JL), Heilongjiang (HL), Inner Mongolia (IM), Shanxi (SX), Hebei (HE), Shandong (SD), Shaanxi (SN), Ningxia (NX), Gansu (GS), Qinghai (QH), Xinjiang (XJ), Tibet (XZ), Sichuan (SC), Yunnan (YN), Guangxi (GX), Guizhou (GZ), Hunan (HN), Guangdong (GD), Fujian (FJ), Zhejiang (ZJ), Jiangxi (JX), Anhui (AH), Jiangsu (JS), Hubei (HB), Henan (HA), Hainan (HI).

The optimal configuration in each province is shown in Fig. 4. Producing 1 kg of hydrogen per hour requires 54.3 kW electrolyzer. The 18 provinces or regions above that are suitable for photovoltaic configuration, with capacities ranging from 80.2 kW (Shanxi) to 107.0 kW (Guangdong). Compared with WE1 and WE2, the costs of WE3 and WE4 are increased. The LCOH of the WE3 is 5.9 $/kg H₂-10.1 $/kg H₂. To ensure 50% of renewable energy penetration, the WE3 needs to be equipped with photovoltaic panels of 143.4 kW-332.1 kW and electrolyzer of 80.0 kW-142.4 kW, with electricity energy storage of 22.3 kWh-413.7 kWh, and hydrogen storage tanks of 4.9 kg-19.1 kg. This has led to an increase in the proportion of investment costs, accounting for 29.1%-42.7%. Accordingly, operation and maintenance costs are increased, and the proportion of power purchase costs is decreased to 20.8%-43.5%. The LCOH is the highest for WE4, 7.3 $/kg H₂-14.8 $/kg H₂, this is because that it is completely off the grid and has a large number of photovoltaic panels (258.8 kW-601.9 kW) and electrolyzer (157.0 kW-283.7 kW) installed. To ensure the reliability of hydrogen supply hourly, larger capacity electricity energy storage (83.5 kWh-939.8 kWh) and hydrogen storage tanks (23.0 kg-137.7 kg) are configured. At the same time, the proportion of investment costs and operation and maintenance costs increase to 52.5%-56.3% and 43.6%-47.4%. In addition, compared to WE1 and WE2, the full load operation hours of the electrolyzer in WE3 and

WE4 are substantially lower. In the case of Beijing, the annual full load operation hours in WE3 and WE4 are only 1679 h and 1107 h respectively, and the full load operation rates are 19.2% and 12.6%, the degradation rates are 2.2% and 1.9%, with only a small impact on the lifetime, as shown in Supplementary Note 3. The degradation rates for constant and dynamic power operation at the same capacity are both 1.9%, using WE4 as an example. If the alkaline (ALK) electrolyzer is replaced, the degradation rates for the same operation condition would be 14.3% and 15.9%. The dynamic power operation has less effect on proton exchange membrane (PEM) electrolyzer but increases the degradation rate of ALK electrolyzer. And the rationale of adopting the PEM electrolyzer is validated.

The spatial distribution of LCCE for the four systems are shown in Fig. 5. The LCCE of WE1, WE2, and WE3 are higher in northern China. This is mainly due to the high carbon emission factors of the grids in northern provinces. For example, the carbon emission factors of Hebei, Inner Mongolia, and Liaoning are 1.092 kg/kWh, 1 kg/kWh, 0.91 kg/kWh, respectively. The LCCE of WE4 are very small on a national scale, which is due to the fact that it is completely off-grid and the carbon emissions only come from photovoltaic panels manufacturing process, with a maximum of only 2.37 kg CO₂/kg H₂. Compared with the CG, the LCCE reduction rate of WE1 is −161.9-77.2%, the

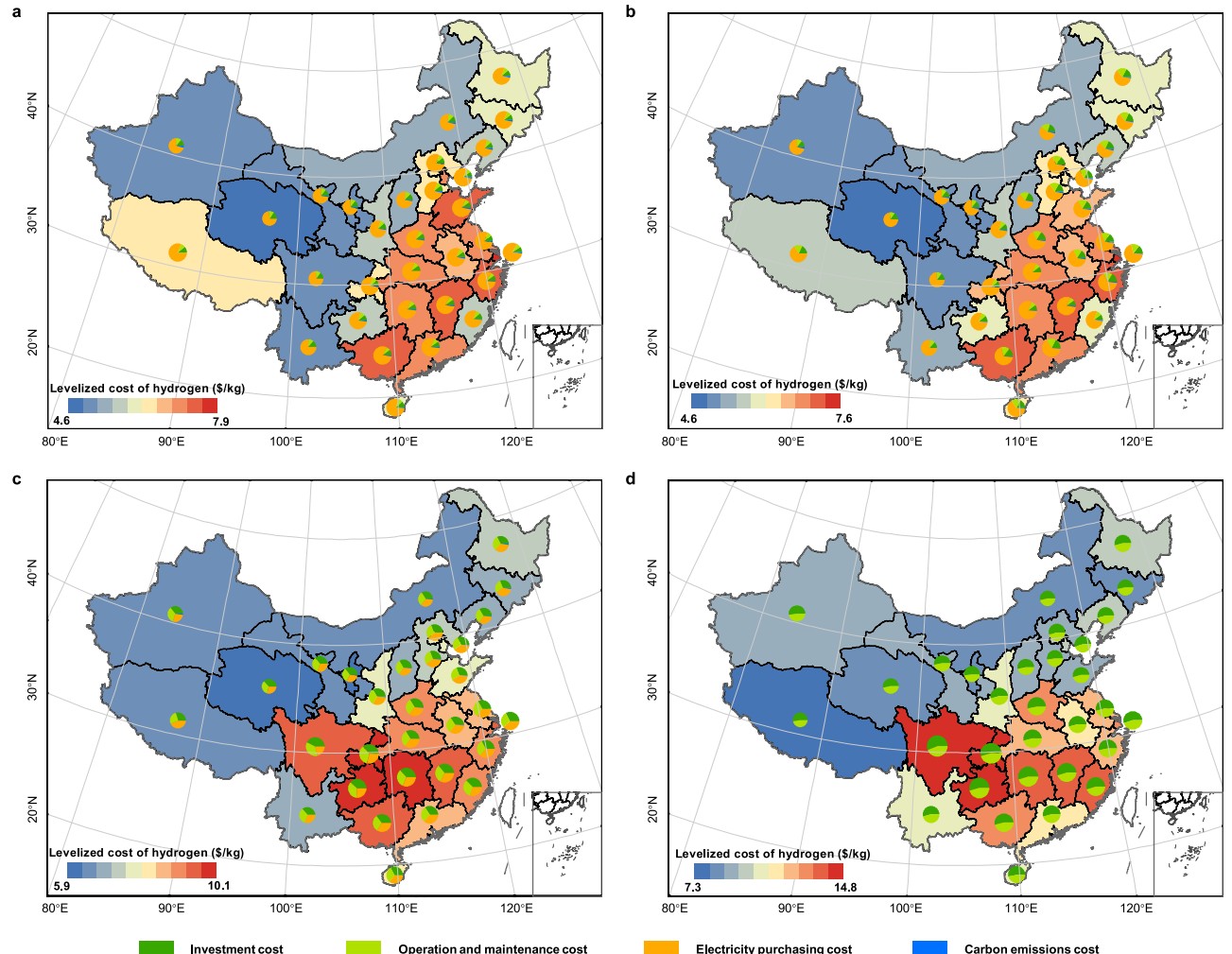

**Fig. 3 | Spatial distribution of levelized cost of hydrogen for four water electrolysis hydrogen production systems. a–d** Panels are WE1, WE2, WE3, WE4, respectively. The shading on the map represents the levelized cost of hydrogen, the pie chart represents the cost components. Map Source: Standard Map Service of the Ministry of Natural Resources, People's Republic of China [Map Review Number: GS (2020) 4619].

LCCE reduction rate of WE2 is −109.1%-81.1%. The LCCE of the WE3 are 3.7 kg $CO_2$/kg $H_2$ (Hebei)-30.8 kg $CO_2$/kg $H_2$ (Qinghai), and the LCCE reduction rate of WE3 increases to −35.9%-83.6%. Except for 8 provinces or regions with high grid carbon emission factors, such as Tianjin, Liaoning, Jilin, Heilongjiang, Inner Mongolia, Shanxi, Hebei, and Ningxia, the LCCE of the rest of the provinces are less than that of the CG. The LCCE reduction rate of WE4 in 31 provinces exceeds 89.5%.

### The impact of time-of-use electricity price on the system

In China, there are two types of electricity prices in the electricity market: single electricity price and time-of-use electricity price. The time-of-use electricity price in each province are shown in Supplementary Fig. 3. Figure 6 presents the impact of applying time-of-use electricity prices on the LCOH of the water electrolysis hydrogen production. Overall, compared with single electricity price, time-of-use electricity prices can reduce the LCOH of the grid-connected water electrolysis hydrogen production in various provinces in China. However, the degree of reduction of time-of-use electricity prices to the LCOH is different for different systems and provinces. Figure 6a–c illustrates the comparison of WE1, WE2, and WE3 applying the single tariff and the time-of-use tariff, respectively, and are arranged in descending order of the reduction in the LCOH, so the order of horizontal axes is not same in the three figures.

For the WE1, the application of time-of-use electricity tariff reduces the LCOH by 0.18 \$/kg $H_2$-0.36 \$/kg $H_2$. Provinces with higher

LCOH, such as Shanghai, Zhejiang, Hunan, Tianjin and Guangdong, showed the highest reduction in LCOH after the application of time-of-use electricity. This is because the cost of purchasing electricity accounts for a large proportion of the cost composition of the WE1 system, which is more sensitive to the change of the price of electricity. For the WE2, the LCOH is reduced by 0.18 \$/kg $H_2$-0.58 \$/kg $H_2$ after time-of-use electricity prices is adopted. 22 provinces are suitable for photovoltaic panels deployment due to time-of-use electricity prices (18 provinces under a single electricity price). Only 9 provinces or regions, Qinghai, Xinjiang, Sichuan, Gansu, Yunnan, Ningxia, Guizhou, Fujian, and Chongqing are more economical with grid-individual hydrogen production. And the figure shows that these 9 provinces have the lowest reduction of LCOH. Therefore, the time-of-use electricity price has a greater impact on the hydrogen production system combined photovoltaic and grid compared to grid hydrogen production system. This is because the adoption of time-of-use electricity prices leads to changes in the installed capacity of photovoltaic panels. The optimal configuration of WE2 under time-of-use electricity prices is shown in Supplementary Fig. 4. For the WE3, the LCOH is reduced by 0.35 \$/kg $H_2$-0.90 \$/kg $H_2$ after adopting time-of-use electricity prices. The provinces or regions, including Tianjin, Xizang, Henan, Guangdong and Hunan, decrease the most. Compared with WE1 and WE2, WE3 is equipped with electricity energy storage and hydrogen storage tank, which can reduce the demand for electricity purchase during

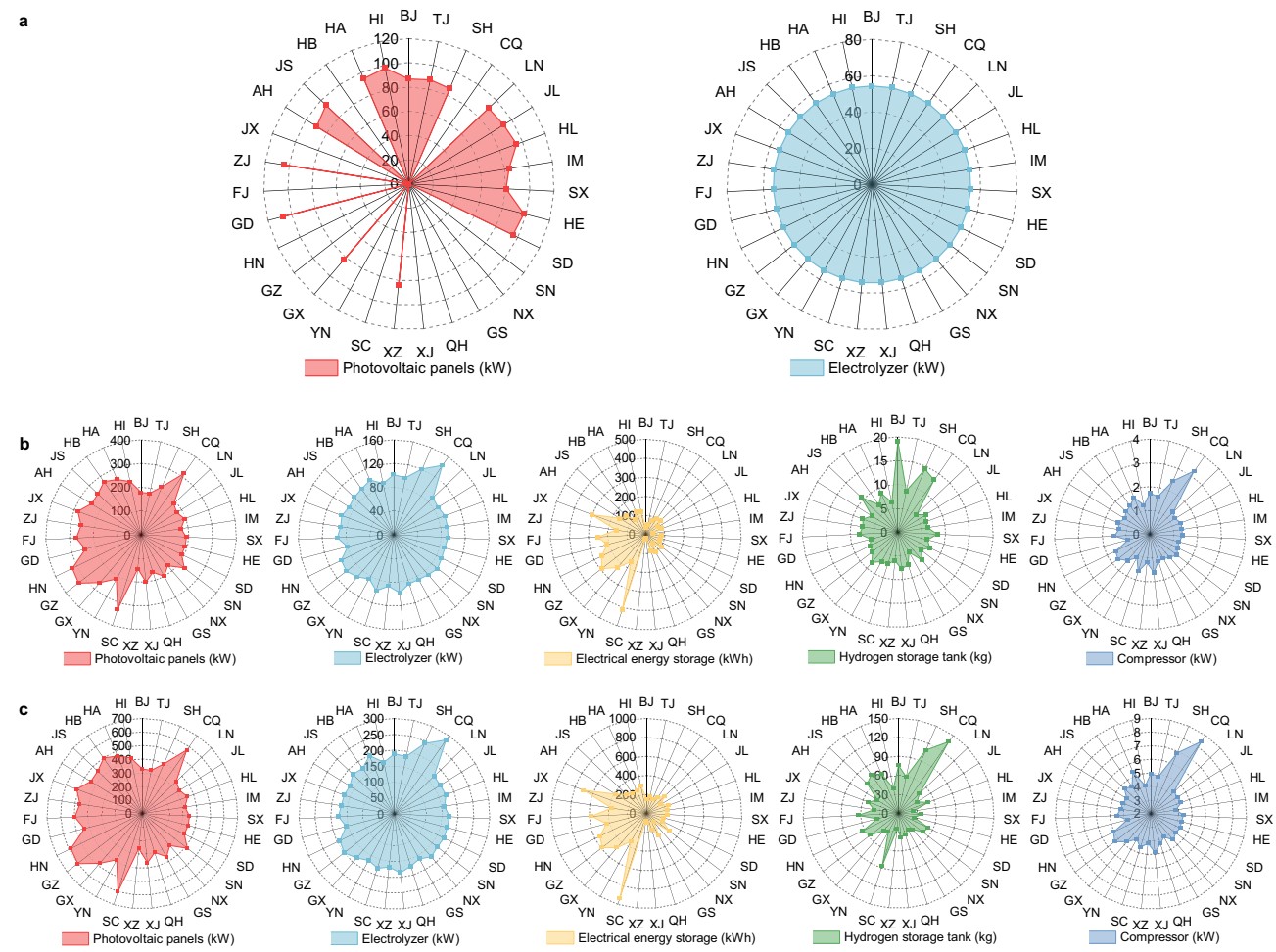

**Fig. 4 | The optimal configuration of water electrolysis hydrogen production systems in each province. a–c** Panels are the optimal configuration for WE2, WE3, and WE4, respectively.

peak electricity price periods and adjust the supply of electricity and hydrogen energy more economically. The optimal configuration is shown in Supplementary Fig. 5. Compared with the single electricity price, the capacities of the electrolyzer and electricity energy storage are changed considerably. Among them, the capacity of the electrolyzer is decreased and the capacity of electricity energy storage is increased.

## Hydrogen production structural transformation considering demand

According to the National Energy Administration of China, the total demand for hydrogen in China is currently 33Mt. The total hydrogen demand is divided according to gross domestic product (GDP), population, and carbon emissions[25]. The hydrogen demand in each province is presented in Supplementary Fig. 6. Hydrogen demand in the provinces ranged from 0.05 Mt to 2.78 Mt. In the future, the share of hydrogen production from renewable energy sources is expected to gradually increase, replacing some of the hydrogen production from fossil energy sources. Therefore, the investment and carbon reduction of transforming from the current to the future hydrogen production structure is quantified, considering the actual demand for hydrogen in each province. The future hydrogen production structure could have a 15% share of hydrogen produced from renewable energy sources, which is predicted by the China Hydrogen Alliance for the year 2030[26]. It is assumed that all provinces are required to complete the transition, and the WE3 and WE4 with lower carbon emissions are representative of hydrogen production from renewable energy sources.

A total of 46.1 billion $ of investment is required for China to achieve a hydrogen production structure transformation to 15% renewable hydrogen, which cumulatively reduces carbon emissions by 61.9 Mt, with a carbon emission reduction rate of 10.1%. The investment and carbon reduction in each province is shown in Fig. 7. Provinces with high hydrogen demand require more investment for the transformation, while reducing more carbon emissions. The largest hydrogen demand is in Guangdong, Jiangsu, and Shandong, with 2.78 Mt, 2.59 Mt, and 2.53 Mt, respectively, and the investment required for the structural transformation of hydrogen production is 3.9 billion $, 3.7 billion $, and 3.1 billion $, respectively, which reduces carbon emissions by 6.3 Mt, 4.5 Mt, and 4.2 Mt, respectively. It is worth noting that although the emission reductions in the above provinces are the largest, the reduction rates are not the most advantageous, at 12.4%, 9.3%, and 8.8%, respectively. Higher carbon reduction rate could be achieved in provinces with abundant photovoltaic resources and low carbon emissions from the grid, such as Qinghai and Tibet, both at 17.0%, however, the hydrogen demand in these provinces is generally low, at 0.13 Mt and 0.05 Mt, respectively.

## Development pathway in future scenarios from 2025 to 2050

In this section, the average LCOH and LCCE of the water electrolysis hydrogen production system of China is calculated by nationwide average data (photovoltaic power generation, electricity price, grid carbon emission factors, etc.). The development pathway of water electrolysis hydrogen production in China during 2025–2050 is further analyzed. The relevant economic and technical parameters of hydrogen production pathways for 2025–2050 are presented in

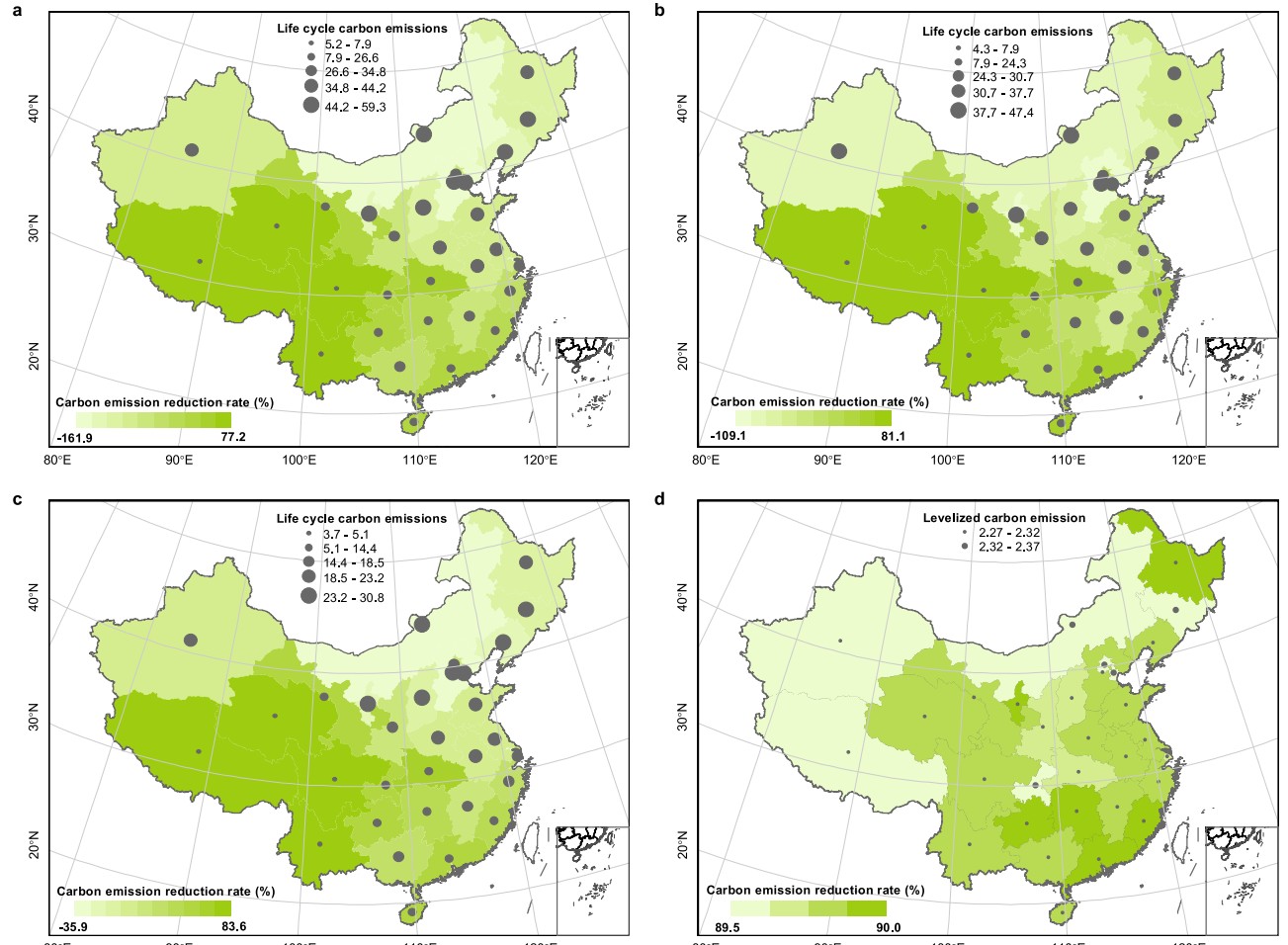

**Fig. 5 | Spatial distribution of life cycle carbon emissions for four water electrolysis hydrogen production systems. a–d** Panels are WE1, WE2, WE3, WE4, respectively. The gray circular symbols represent life cycle carbon emissions (unit: kg CO₂/kg H₂). The green shading on the map represents the carbon emission reduction rate of water electrolysis hydrogen production systems relative to coal gasification hydrogen production in each province. Map Source: Standard Map Service of the Ministry of Natural Resources, People's Republic of China [Map Review Number: GS (2020) 4619].

Supplementary Note 6. A total of four scenarios are set up. The first scenario is the baseline scenario, aiming to minimize the LCOH. On this basis, the other three scenarios consider subsidy policies for water electrolysis hydrogen production, which are derived from actual policies currently being implemented in Chinese provinces and cities, as shown in Methods. Among them, 50% of the electricity price concession is considered in scenario 2. The incentives for carbon reduction are considered in scenario 3, which means that 50% of the project investment are subsidized when the carbon emission of water electrolysis hydrogen production is less than that of CG and SMR. Scenario 4 is a direct subsidy of 50% on the hydrogen production cost.

The tendency of the cost and low-carbon competition of water electrolysis hydrogen production under the baseline scenario is shown in Fig. 8a, b. With the reduction of the investment cost of the equipment related to hydrogen production system and the improvement of the efficiency of the electrolyzer, the LCOH of four water electrolysis hydrogen production systems are decreased year by year. Specifically, the four systems are decreased by 2.2 $/kg H₂, 2.8 $/kg H₂, 3.9 $/kg H₂, 6.8 $/kg H₂, respectively. With the increase of carbon emission cost, the LCOH of hydrogen production from coal and natural gas increased year by year. Specifically, CG, CG + CCUS, SMR, SMR + CCUS increases by 1.2 $/kg H₂, 0.8 $/kg H₂, 0.8 $/kg H₂ and 0.5 $/kg H₂ respectively. It is conspicuous that the LCOH of WE4 could be reduced to below the LCOH of WE1, WE2, and WE3 in 2035, which means that off-grid hydrogen production is likely to have economic and environmental

advantages, and may become the mainstream way of water electrolysis hydrogen production after 2035. In 2045, the LCOH of WE4 could be reduced to 2.8 $/kg H₂, lower than the LCOH of CG + CCUS, as shown in the first key point of Fig. 7. The LCOH of WE4 could be reduced to 2.2 $/kg H₂ in 2050, and lower than industry by-product. As shown in the second key point, WE4 could become the most economical way to produce hydrogen. In addition, as the proportion of renewable energy in the grid increases, the carbon emission factor of the grid decreases year by year, and the LCCE of WE1, WE2 and WE3 are reduced. Specifically, the reduction is 22.5 kg CO₂/kg H₂, 16.8 kg CO₂/kg H₂ and 15.5 kg CO₂/kg H₂, respectively. It is worth noting that the carbon emissions of WE3, WE2, and WE1 could be reduced below other hydrogen production ways in 2035, 2040, and 2045, respectively.

The tendency of the cost competition of water electrolysis hydrogen production under the scenario 2, 3, and 4 is shown in Fig. 8c–e. The subsidy policy is conducive to driving down the LCOH of water electrolysis hydrogen production and accelerating the progress towards the time when they are cost-competitive. Compared to the baseline scenario, the first key point could be advanced by the three subsidy policies, about 5 years, 5 years, and 15 years, respectively. Under carbon reduction incentives and production subsidies, the second key point could be advanced by ~5 years and 15 years, respectively, i.e., the off-grid water electrolysis hydrogen production system could become the most economical and low-carbon hydrogen production pathway much earlier. In addition, the electricity price subsidy

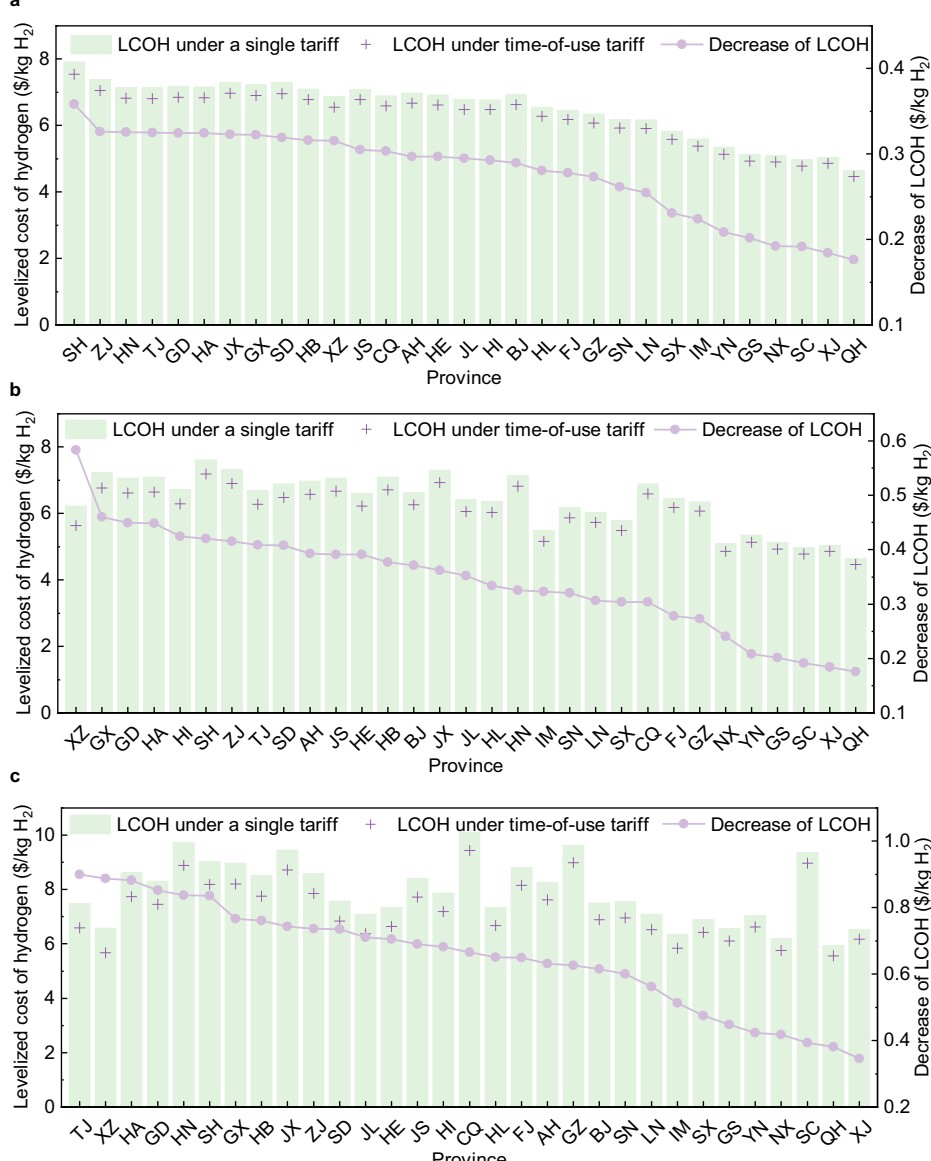

**Fig. 6 | The impact of time-of-use electricity prices on the levelized cost of hydrogen for water electrolysis hydrogen production systems. a–c** Panels are WE1, WE2, WE3, respectively. The green bar labels represent the levelized cost of hydrogen (LCOH) under a single electricity tariff. The purple symbols represent the levelized cost of hydrogen (LCOH) under time-of-use electricity tariff. The purple dot symbols represent the difference between the levelized cost of hydrogen (LCOH) under the two electricity tariffs.

policy may have no effect on WE4. Because the investment cost accounts for a small proportion, the investment incentive policy for reducing carbon has a small impact on WE1 and WE2. The production subsidy could simultaneously promote the reduction of the LCOH of four water electrolysis hydrogen production systems. This suggests that different policies are applicable to different water electrolysis hydrogen production systems, targeted policies should be issued and promoted by each province according to their natural conditions and the type of water electrolysis hydrogen production which is suitable for development.

## Discussion

In view of the differences in renewable resources, electricity price levels and types, and grid carbon emission among Chinese provinces, the LCOH and LCCE for four water electrolysis hydrogen production systems are quantified in 31 provinces and regions in China, and compares with hydrogen production from coal, natural gas, and

industry by-product. The development path of water electrolysis hydrogen production in future scenarios are explored.

The research findings indicate that the LCOH for water electrolysis hydrogen production is generally high in the eastern and southern coastal areas of China, and is relatively low in western and northern regions. The high carbon emission factor of the grid in northern China leads to the high LCCE of grid-connected water electrolysis hydrogen production system. In 18 provinces or regions including Beijing, Tianjin, Shanghai, Liaoning, Jilin, Heilongjiang, Inner Mongolia, Shanxi, Hebei, Shandong, Tibet, Guangxi, Guangdong, Zhejiang, Anhui, Jiangsu, Henan, and Hainan, currently water electrolysis hydrogen production can be developed by combining photovoltaic and grid, as it is more economical and environmentally friendly than the grid hydrogen production system. The remaining 13 provinces and regions are more economical with grid-alone hydrogen production due to low electricity prices and small generation of photovoltaic power. For example, the annual photovoltaic power generation and electricity price of Sichuan province are 0.75

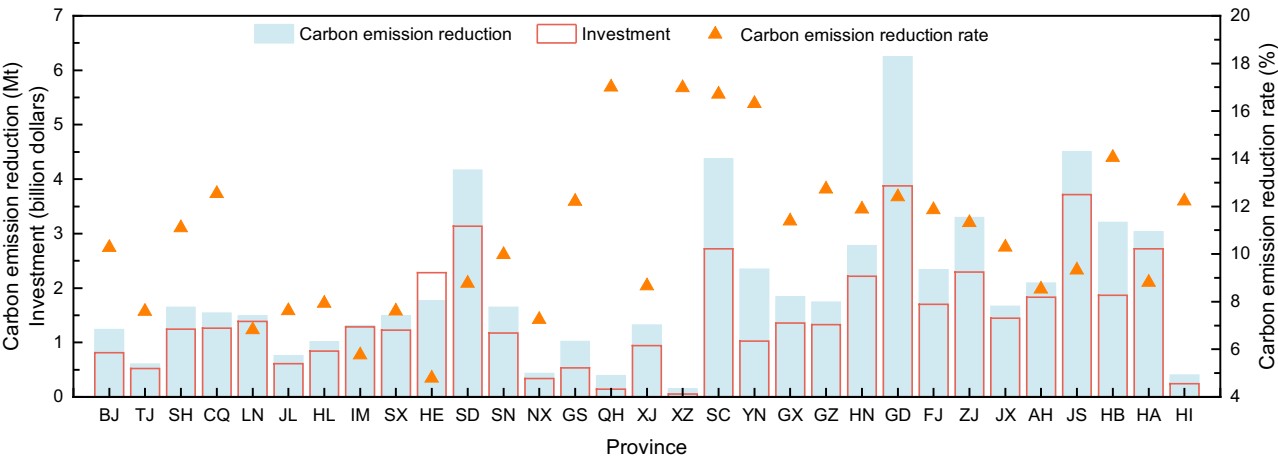

**Fig. 7 | Investment and carbon emission reduction for hydrogen production structural transformation in provinces of China.** The blue color represents the carbon emission reductions of the hydrogen production structural transformation, the red frame represents the required investment, and the yellow triangle represents the carbon emission reduction rates.

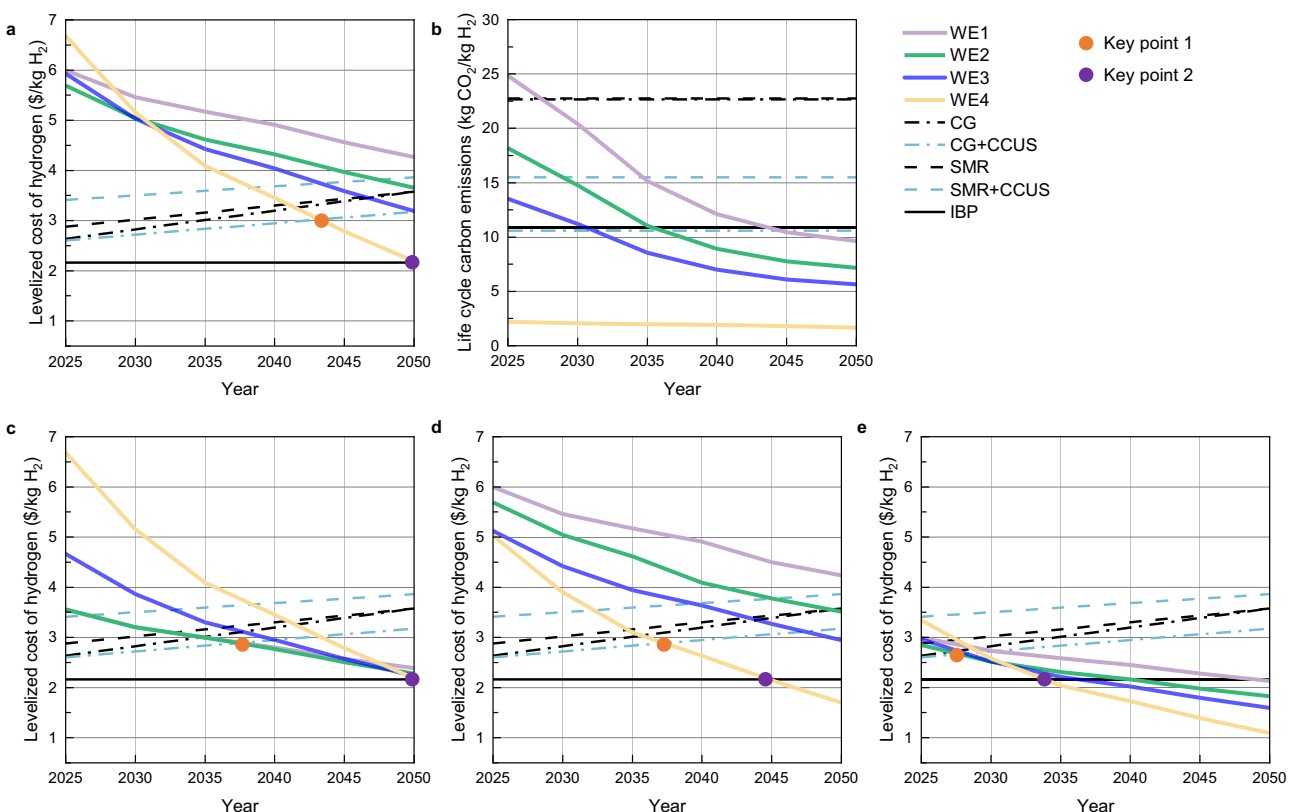

**Fig. 8 | Development pathways for water electrolysis hydrogen production.** **a**, **b** Panels are the levelized cost of hydrogen and life cycle carbon emissions, respectively, under scenario 1. **c**–**e** Panels are the levelized cost of hydrogen under scenarios 2, scenario 3, and scenario 4, respectively. Key point 1 is the point where the cost of water electrolysis hydrogen production is lower than that of hydrogen production from fossil fuels. Key point 2 is the point where water electrolysis hydrogen production becomes the cheapest pathway to produce hydrogen.

MWh/year and 0.056 $/kWh respectively. The renewable energy penetration increases sequentially in WE1, WE1, WE3 and WE4, resulting in an increase in LCOH and a decrease in LCCE, thus if environmental protection performance is a priority when planning a water electrolysis hydrogen production project, WE3 and WE4 need to be deployed at the expense of economy. The configuration schemes in Fig. 4 could help hydrogen project investors to deploy water electrolysis hydrogen production systems in the provinces at the most economical way possible.

In addition, compared to other hydrogen production pathways, the LCOH of water electrolysis hydrogen production is the highest in the provinces, which is the key challenge that currently constrains its development. For grid-connected water electrolysis hydrogen production, the key to reducing LCOH is to decrease the cost of purchased electricity, which accounts for a large portion of costs. The time-of-use electricity price can be promoted expeditiously to enhance the economic competitiveness of such systems, especially in provinces with high

electricity price, such as Shanghai and Zhejiang, etc. And grid-connected water electrolysis hydrogen production has low-carbon advantages only in some provinces, such as Qinghai, Tibet, Sichuan, and Yunnan, so the system could be prioritized for development in these provinces, similar to the conclusions drawn in Literature[27]. For off-grid water electrolysis hydrogen production system, it can achieve low carbon emissions in all provinces at present although substantial investment is brought by configuring larger capacities for electricity energy storage and hydrogen storage tank. It could have both economic and low-carbon advantages during 2024–2050, certain carbon reduction incentives or production subsidies may enable this to happen 5–15 years earlier. Before it becomes economically advantageous, hydrogen production from industry by-product, which has lower cost and carbon emission, is a good alternative to water electrolysis hydrogen production.

These key findings provide important insights for China's hydrogen strategy deployment, suggesting a regionally differentiated and temporally phased approach. In the near term, hydrogen production through industry by-products should be prioritized to establish market foundations, particularly in eastern coastal provinces with mature industrial infrastructure and substantial hydrogen demand, such as Shandong, Jiangsu, and Zhejiang. Grid-connected water electrolysis hydrogen production systems could be strategically deployed in regions like Qinghai, Tibet, Sichuan, and Yunnan, while simultaneously initiating demonstrations of off-grid water electrolysis systems. In the long term, off-grid water electrolysis systems could be deployed nationwide, with targeted subsidies provided in regions demonstrating strong economic viability. This strategic approach not only minimizes short-term implementation costs and hydrogen-use carbon emissions but also indirectly expands renewable energy share to ensure long-term sustainability, thereby helping the country to meet its 2060 carbon neutrality target.

Despite these findings, there are a number of limitations that should be acknowledged. The optimization and evaluation are conducted for different types of water electrolysis hydrogen production systems, but the uncertainty of photovoltaic panels for power generation is not considered in the optimization model, as well as the limitation of raw materials and land area required for the installation of renewable energy equipment, electrolyzer, and energy storage equipment, which could change the optimal configuration and LCOH of the system in the engineering practice to a certain extent.

The mismatch between the spatial distribution of renewable hydrogen resources and demand can be observed through the distribution of carbon emission reduction and reduction rate in each province. How to solve the above problems of mismatch by utilizing low-cost and long-distance hydrogen transmission technologies is the focus of future study.

## Methods
### Overview of hydrogen production pathways
Each hydrogen production pathway needs to achieve the same hydrogen supply reliability (hydrogen supply rate) for a fair comparison of LCOH. More than 90% of the global hydrogen used in industrial applications such as refining, ammonia synthesis, and methanol preparation, a stable supply of hydrogen is an absolute requirement for safe industrial production[12]. Most hydrogen production equipment is currently arranged in industrial plants with staggered downtime for maintenance to guarantee a continuous and stable supply of hydrogen[1,28]. For these reasons, it is necessary and important to limit the hourly hydrogen supply for each hydrogen production pathway to a fixed value[29]. In this paper, it is assumed that the hydrogen supply rate is 1 kg/h. A sensitivity analysis is conducted for hydrogen supply rates, which are set at 1 kg/h, 10 kg/h, 100 kg/h, 1000 kg/h, and 10000 kg/h, respectively. The results are shown in Supplementary Fig. 7, which indicates that LCOH does not change with hydrogen supply rate. Therefore, this assumption is valid.

### Water electrolysis hydrogen production systems
System 1 (WE1): the grid hydrogen production system. In this system, all the electric power of the electrolyzer comes from the grid, and the capacity of the electrolyzer can be obtained directly through calculation, i.e., a 54.3 kW electrolyzer is required for the production per hour of 1 kg hydrogen.

System 2 (WE2): the hydrogen production system combined photovoltaic and grid. In this system, the electrical power of the electrolyzer comes from photovoltaics or the grid. The main motivation for this configuration is that this system can integrate distributed photovoltaics. Modular distributed photovoltaic technology makes hydrogen production systems easier to deploy near hydrogen applications[30,31]. The grid is used as a backup power source, which ensures the hourly reliability of the hydrogen supply, thus eliminating the need to equip compression and energy storage[32]. It is worth noting that photovoltaic panels are not mandatory to install, depending on the results of optimization.

System 3 (WE3): the hydrogen production system with fixed renewable energy penetration. In this system, the renewable energy penetration is set at 50%. This constraint is added to reduce the carbon emissions generated from the water electrolysis hydrogen production purchasing electricity from the grid. The electricity for the electrolyzer comes from the grid and photovoltaic panels, and the supply of electricity is regulated by the electricity energy storage, and the supply of hydrogen is regulated by the hydrogen storage tank. A stable supply of hydrogen on an hourly basis can be ensured through the combination of these two energy storage methods[29]. It is worth noting that electricity energy storage and hydrogen storage tank are not necessarily installed in the system, this depends on the optimization results.

System 4 (WE4): the off-grid hydrogen production system. The motivation for this configuration is twofold. On the one hand, this system can achieve 100% renewable energy penetration and near-zero carbon emissions for hydrogen production[33]. On the other hand, some geographical islands and other remote areas cannot access the national grid[34]. Electricity energy storage and hydrogen storage tank play a role in peak cutting and valley filling, ensuring a stable and reliable supply of hydrogen. Whether electricity energy storage and hydrogen storage tank are configured depends on the optimization results.

The four water electrolysis hydrogen production systems have different renewable energy penetrations. The systems all use PEM electrolyzer to produce hydrogen by electrolyzing water, which has the advantages of green and flexible production, and high purity[35].

### Hydrogen production from coal
Hydrogen production from coal is currently the most widely used hydrogen production technology in China. It is mainly composed of CG, which converts coal into syngas through gasification technology, and then goes through water-to-coal gas conversion and separation to extract high-purity hydrogen[36]. Currently, the pathway of hydrogen production from coal is relatively mature and can be prepared on a large scale in a stable manner[37]. In this research, it is assumed that the energy input to the coal gasifier is only coal, hydrogen is delivered directly to consumers once it is produced. The reliability of hydrogen is ensured at a constant supply rate per hour, which is necessary for hydrogen using plants. In addition, on the basis of this system, the CG + CCUS is also considered, which changes the investment cost, conversion efficiency, and carbon emission coefficient of the system.

### Hydrogen production from natural gas
The primary method of hydrogen production from natural gas is SMR, where the energy inputs include grid electricity and natural gas[38]. Once hydrogen is produced, it is delivered directly to consumers. The SMR + CCUS is also constructed in this research. The investment cost, operational cost, electricity consumption, natural gas consumption, and direct carbon emission coefficient of this system are changed following the integration of CCUS[39].

### Hydrogen production from industry by-products

Hydrogen-rich industrial exhaust gases usually are regarded as raw materials in hydrogen production from industry by-product. Hydrogen with a purity of over 99.9% can be gained after separating and purifying the hydrogen gas through techniques such as pressure swing adsorption. Currently, the main sources include industry by-product from the chlor-alkali industry, coking-oven gas, and light hydrocarbon cracking by-product gas[40]. Hydrogen production from industry by-product requires almost no additional capital investment and fossil fuel inputs compared to other hydrogen production pathways, which is the greatest advantage Moreover, China has unique advantages due to its large number of industries.

### Hydrogen production system optimization model

The hourly power generation data of photovoltaic panels with a rated power of 1 kW are collected from the PVWatts Calculator developed by the National Renewable Energy Laboratory, USA[41]. The electricity price comes from the electricity price sales table of each provincial power grid company. The above data and the carbon emission factors of each provincial power grid and the technical-economic parameters of the hydrogen production system equipment are inputted into the hydrogen production system optimization model, and the model framework is presented in Fig. 9. An optimization model for the configuration and full-year hourly operation of the water electrolysis hydrogen production system is established, considering multiple constraints such as equipment operation, power purchases, electricity balance, hydrogen balance, and hydrogen supply reliability. Through this model, the investment and operation schemes of multiple water electrolysis hydrogen production systems in different provinces for China can be jointly optimized under different subsidy policies for different target years, while the LCCE of the systems can be assessed. The hydrogen production system optimization model is constructed as a mixed-integer linear programming (MILP) model. MILP, which efficiently solve linear equations and ensure the balance between computational efficiency and robustness, has become a predominant optimization method for the design and operation of energy systems and has been widely used. Based on the Matlab platform, the Gurobi solver is invoked through the Yalmip toolbox to solve the model.

The optimization model for the water electrolysis hydrogen production system, such as WE3, is as follows. The code is available in the Supplementary Software. Optimization objectives, decision variables, and constraints are the three necessary parts for the optimization model. The optimization models for WE1, WE2, and WE4 are provided in Supplementary Note 1. The parameters and data involved in the optimization model are listed in Supplementary Note 2.

### Water electrolysis hydrogen production optimization model

The LCOH is an internationally recognized economic evaluation indicator for the hydrogen energy industry. It represents the levelized monetary cost required to produce 1 kg of hydrogen throughout full life cycle of a hydrogen production project. A lower LCOH indicates lower hydrogen production costs for this technology pathway, and higher market competitiveness[5]. The LCOH for WE3 can be calculated as follows:

$$LCOH_{WE3} = \frac{C_{inv,WE3} + C_{om,WE3} + C_{grid,WE3} + C_{c,WE3}}{\sum_{n=1}^{N} \frac{H_{output}}{(1+r)^n}} \tag{1}$$

$$C_{inv,WE3} = c_{inv,PV} \cdot IC_{PV} + c_{inv,EL} \cdot IC_{EL} + c_{inv,EES} \cdot IC_{EES} + c_{inv,HST} \cdot IC_{HST} + c_{inv,com} \cdot IC_{com} \tag{2}$$

$$C_{om,WE3} = \sum_{n=1}^{N} \frac{c_{om,PV} \cdot IC_{PV} + c_{om,EL} \cdot IC_{EL} + c_{om,EES} \cdot IC_{EES} + c_{om,HST} \cdot IC_{HST} + c_{om,com} \cdot IC_{com}}{(1+r)^n} \tag{3}$$

$$C_{grid,WE3} = \sum_{n=1}^{N} \frac{\sum_{t=1}^{t=h}(c_{grid} \cdot EP_{grid}(t) + c_{grid,fix1} \cdot EP_{grid,max} + \frac{c_{grid,fix2} \cdot EP_{grid,max}}{\cos\varphi}}{(1+r)^n} \tag{4}$$

$$C_{c,WE3} = c_c \cdot LCCE_{WE3} \tag{5}$$

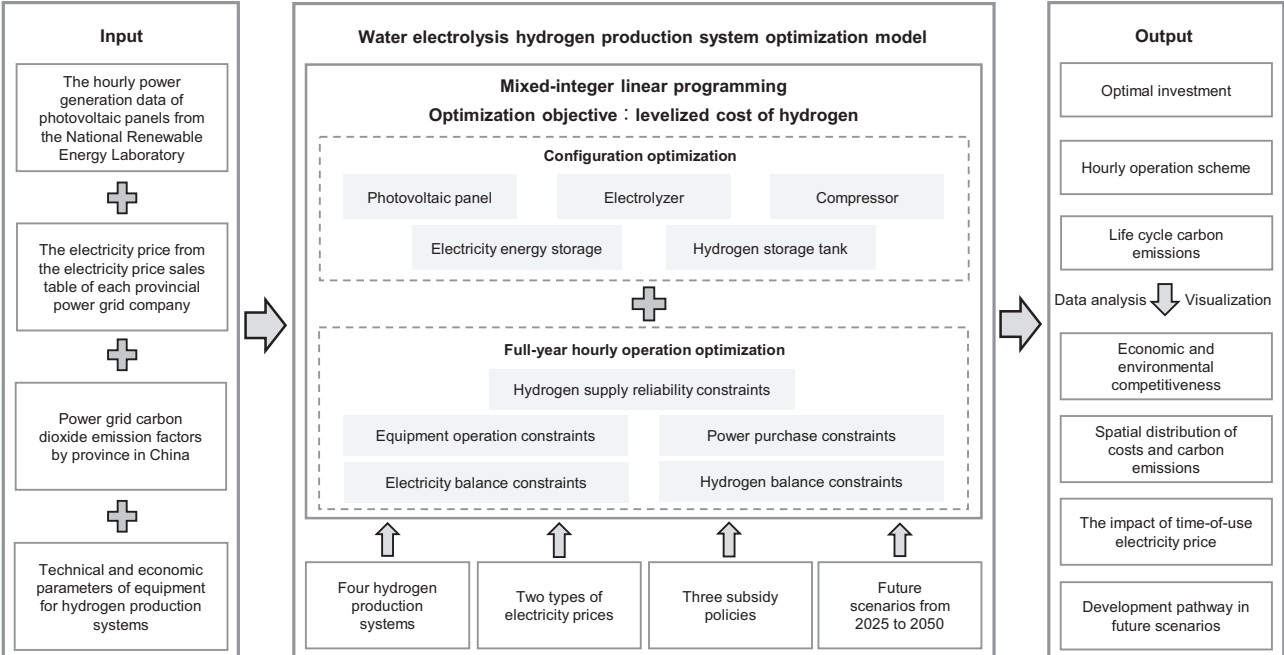

**Fig. 9 | Modeling framework for hydrogen production system optimization.** The optimization framework contains inputs, optimization models, and outputs. The optimization model is mixed-integer linear programming, the optimization objective is to minimize the levelized cost of hydrogen, and the decision variables include configuration and full-year hourly operation.

where $LCOH_{WE3}$ is the LCOH of the water electrolysis hydrogen production system, $C_{inv,WE3}$, $C_{om,WE3}$, $C_{grid,WE3}$ and $C_{c,WE3}$ are respectively the cost of investment, the cost of operation and maintenance throughout full life cycle, the cost of power purchase throughout full life cycle, and the cost of carbon emissions throughout full life cycle of the system. $r$ is discount rate, %. $N$ is life, years. $c_{inv,PV}$, $c_{inv,EL}$, $c_{inv,EES}$, $c_{inv,HST}$, $c_{inv,com}$ are the cost of unit investment of photovoltaic panels, electrolyzer, electricity energy storage, hydrogen storage tanks, and compressors respectively. $c_{om,PV}$, $c_{om,EL}$, $c_{om,EES}$, $c_{om,HST}$, $c_{om,com}$ are the costs of unit operation and maintenance of photovoltaic panels, electrolyzer, electricity energy storage, hydrogen storage tanks, and compressors respectively. The technical and economic parameters of each equipment are shown in Supplementary Table 1. $c_{grid}$ is the purchase price of electricity, $/kW. $c_{grid,fix1}$ and $c_{grid,fix2}$ are the maximum demand price and transformer capacity price respectively, $/kW-year and $/kVA-year, $\varphi$ is the phase angle. The retail electricity price of 35 kV large-scale industrial electricity is used as the representative electricity price for purchasing electricity from the grid. The electricity prices in each province are shown in Supplementary Table 2. $c_c$ is the cost of carbon emission, $/t; $LCCE_{WE3}$ is the LCCE of the WE3.

Decision variables include system configuration and operation. The decision variables of configuration include the capacity of photovoltaic panels ($IC_{PV}$, kW), the capacity of electrolyzer ($IC_{EL}$, kW), the capacity of compressor ($IC_{com}$, kW), and the capacity of electricity energy storage ($IC_{EES}$, kWh), the capacity of the hydrogen storage tank ($IC_{HST}$, kg). The decision variables of operation include the input power of the electrolyzer ($EP_{EL}(t)$, kW), the production of hydrogen of the electrolyzer ($H_{EL}(t)$, kg), the power consumption of the compressor ($EP_{com}(t)$, kW), the charging power of the electricity energy storage ($EP_{EES,im}(t)$, kW), the discharging power of the electricity energy storage ($EP_{EES,ex}(t)$, kW), hydrogen storage of the hydrogen storage tank ($H_{HST,im}(t)$, kg), hydrogen release of the hydrogen storage tank ($H_{HST,ex}(t)$, kg), power purchased from the grid ($EP_{grid}(t)$, kW).

**Equipment operation constraints.** The annual photovoltaic power generation data of each province in the country is shown in Supplementary Fig. 1. Photovoltaic power generation and photovoltaic capacity are required to meet the following constraints[42,43].

$$EP_{PV}(t) = IC_{PV} \cdot EP_{PV,0}(t) \tag{6}$$

$$0 \le IC_{PV} \tag{7}$$

where $EP_{PV}$ is electric power output of photovoltaic panels, kW. $EP_{PV,0}$ is the output electric power of the photovoltaic panel with a rated capacity of 1 kW. $IC_{PV}$ is the installation capacity of photovoltaic panels, kW.

The electrolyzer generates hydrogen by consuming electricity to electrolyze water. The energy conversion model is shown in Eq. (8). The input power and capacity of the electrolyzer are required to meet the constraints of Eqs. (9) and (10) respectively[27].

$$H_{EL}(t) \cdot LHV = \eta_{EL}(t) \cdot EP_{EL}(t) \tag{8}$$

$$0 \le EP_{EL}(t) \le IC_{EL} \tag{9}$$

$$0 \le IC_{EL} \tag{10}$$

where $EP_{EL}$ is the input electric power, kW. $H_{EL}$ is the production of hydrogen, kg; $IC_{EL}$ is the installation capacity of electrolyzer, kW. $\eta_{EL}$ is the conversion efficiency of the electrolyzer, $LHV$ is the lower calorific value of hydrogen, 33.3 kWh/kg.

The compressor compresses hydrogen to 200 bar and stores it in hydrogen storage tank. When hydrogen storage is needed, the compressor starts. This process is required to meet the following constraints[33].

$$EP_{com}(t) = EP_{com,0} \cdot H_{EL}(t) \cdot LHV \tag{11}$$

$$EP_{com}(t) \le IC_{com} \tag{12}$$

where $EP_{com,0}$ is the electrical energy required to compress 1 kg of hydrogen, kWh. $EP_{com}$ is the hourly electrical energy consumption of the compressor, kW. $IC_{com}$ is the capacity of the compressor, kW.

The lithium battery can be installed to store electricity to ensure a stable power supply for the electrolysis process, as shown in the mathematical model below[44].

$$E_{EES}(t+1) = E_{EES}(t)(1-\alpha) + (EP_{EES,im}(t)\eta_{EES,im} - EP_{EES,ex}(t)/\eta_{EES,ex})\Delta t \tag{13}$$

$$SOC_{EES}(t) = E_{EES}(t)/IC_{EES} \tag{14}$$

$$SOC_{EES,min} \le SOC_{EES}(t) \le SOC_{EES,max} \tag{15}$$

$$0 \le EP_{EES,im}(t) \le EP_{EES,im,max} \tag{16}$$

$$0 \le EP_{EES,ex}(t) \le EP_{EES,ex,max} \tag{17}$$

$$EP_{EES,im}(t) \cdot EP_{EES,ex}(t) = 0 \tag{18}$$

$$SOC_{EES}(t_{start}) = SOC_{EES}(t_{end}) \tag{19}$$

where $E_{EES}(t)$ is the electricity, kWh; $SOC_{EES}(t)$ is the real-time state of charge; $E_{EES}(t+1)$ is the electricity after charging or discharging, kWh; $\alpha$ is the self-discharge rate; $EP_{EES,im}$ and $EP_{EES,ex}$ are charging power and discharging power, respectively, kW; $\eta_{EES,im}$ and $\eta_{EES,ex}$ are charging efficiency and discharging efficiency, respectively; $EP_{EES,im,max}$ and $EP_{EES,ex,max}$ are the maximum charging power and discharging power, respectively; $IC_{EES}$ is the installation capacity, kWh; $SOC_{EES,min}$ and $SOC_{EES,max}$ are the minimum and maximum state of charge respectively. Equation (18) limits the charging and discharging at the same time of the electrical energy storage. Equation (19) ensures that the initial and final state of the electrical energy storage is consistent.

Surplus renewable electricity can also be used to produce hydrogen directly, and the hydrogen produced by the electrolyzer can be compressed and stored in hydrogen storage tanks to ensure the hydrogen supply. The pressure of hydrogen storage tank is 200 bar. The mathematical model of the hydrogen storage tank is as follows[45].

$$m_{HST}(t+1) = m_{HST}(t) + H_{HST,im}(t) \cdot \Delta t - H_{HST,ex}(t) \cdot \Delta t \tag{20}$$

$$LOH_{HST}(t) = m_{HST}(t)/IC_{HST} \tag{21}$$

$$0 \le H_{HST,im} \le H_{HST,im,max} \tag{22}$$

$$0 \le H_{HST,ex} \le H_{HST,ex,max} \tag{23}$$

$$0 \le m_{HST}(t) \le IC_{HST} \tag{24}$$

$$LOH_{\text{HST,mim}} \leq LOH_{\text{HST}}(t) \leq LOH_{\text{HST, max}} \qquad (25)$$

$$H_{\text{HST,im}}(t) \cdot H_{\text{HST,ex}}(t) = 0 \qquad (26)$$

$$LOH_{\text{HST}}(t_{\text{start}}) = LOH_{\text{HST}}(t_{\text{end}}) \qquad (27)$$

where $m_{\text{HST}}$ is the quality of hydrogen, kg. $H_{\text{HST,im}}$ and $H_{\text{HST,ex}}$ are the rates of hydrogen import and export by the hydrogen storage tank, respectively, kg/h. $LOH_{\text{HST}}$ is the level of hydrogen. $H_{\text{HST,im,max}}$ and $H_{\text{HST,ex,max}}$ are the maximum rates of hydrogen import and export, respectively, kg; $IC_{\text{HST}}$ is the installation capacity, kg; $LOH_{\text{HST,min}}$ and $LOH_{\text{HST,max}}$ are the minimum and maximum level of hydrogen; Eq. (26) limits hydrogen import and export at the same time; Eq. (27) ensures that the initial and final state of the hydrogen storage tank is consistent.

**Power purchase constraints.** The system is required to meet the following power purchase constraints to ensure that 50% renewable energy penetration.

$$0 \leq EP_{\text{grid}}(t) \leq EP_{\text{grid, max}} \qquad (28)$$

$$\sum_{t=0}^{t=h} EP_{\text{grid}}(t) = 50\% \cdot \sum_{t=0}^{t=h} EP_{\text{EL}}(t) \qquad (29)$$

where $EP_{\text{grid}}$ is the purchase electricity, kW; $EP_{\text{grid,max}}$ is the upper limit of purchase electricity, kW; $h$ is total hours per year, 8760.

**Electricity balance constraints.** The electricity balance constraint of the system are as follows.

$$EP_{\text{PV}}(t) + EP_{\text{grid}}(t) + EP_{\text{EES,ex}} = EP_{\text{EES,im}} + EP_{\text{EL}}(t) + EP_{\text{com}}(t) \qquad (30)$$

**Hydrogen energy balance constraints.** The hydrogen produced by the electrolyzer can be supplied directly to users or stored through hydrogen storage tank. Therefore, the hydrogen energy balance constraint is as follows.

$$H_{\text{EL}}(t) = H_{\text{EL,out}}(t) + H_{\text{HST,im}}(t) \qquad (31)$$

where $H_{\text{EL,out}}$ is the hydrogen energy provided by the electrolyzer to the hydrogen energy application scenario, kg.

**Hydrogen supply reliability constraints.** Hydrogen supply is required to meet hourly reliability[29].

$$H_{\text{EL,out}}(t) + H_{\text{HST,ex}}(t) = \frac{H_{\text{output}}}{h} \quad t \in 1, 2, \cdots h \qquad (32)$$

where $H_{\text{output}}$ is the annual supply of hydrogen of the system, 8760 kg.

**Hydrogen production from coal optimization model**
The LCOH of CG is calculated as follows[27].

$$LCOH_{\text{CG}} = \frac{c_{\text{inv,CG}} \cdot IC_{\text{CG}} + \sum_{n=1}^{N} \frac{c_{\text{om,CG}} \cdot IC_{\text{CG}} + (c_{\text{coal}} + c_c \cdot \lambda_{\text{CG}} \cdot \eta_{CG}) \sum_{t=1}^{t=h} P_{\text{CG}}(t)}{(1+r)^n}}{\sum_{n=1}^{N} \frac{H_{\text{output}}}{(1+r)^n}} \qquad (33)$$

$$0 \leq P_{\text{CG}}(t) \leq IC_{\text{CG}} \qquad (34)$$

$$0 \leq IC_{\text{CG}} \qquad (35)$$

$$H_{\text{CG}}(t) = \eta_{\text{CG}} \cdot P_{\text{CG}}(t) \qquad (36)$$

$$H_{\text{CG}}(t) = \frac{H_{\text{output}}}{h} \qquad (37)$$

where $LCOH_{\text{CG}}$ is the LCOH of CG. $c_{\text{inv,CG}}$ is the investment cost of CG, \$/kW. $IC_{\text{CG}}$ is the optimal capacity, kW. $c_{\text{om,CG}}$ is the operation and maintenance cost of CG, \$/kW; $c_{\text{coal}}$ is the price of unit coal, 81.5 \$/t. $c_c$ is the emission price of unit carbon dioxide, 4.1 \$/t. $\lambda_{\text{CG}}$ is the $CO_2$ released by the CG per 1 kg of hydrogen produced. $\eta_{\text{CG}}$ is the conversion efficiency of CG. Difference exists on investment cost, conversion efficiency and carbon emission coefficient between CG + CCUS and CG[46]. The technical and economic parameters of the two systems are shown in Supplementary Table 3.

**Hydrogen production from natural gas optimization model**
The LCOH of SMR is calculated as follows.

$$LCOH_{\text{SMR}} = \frac{C_{\text{inv,SMR}} + C_{\text{om,SMR}} + C_{\text{gas,SMR}} + C_{\text{grid,SMR}} + C_{\text{c,SMR}}}{\sum_{n=1}^{N} \frac{H_{\text{output}}}{(1+r)^n}} \qquad (38)$$

$$C_{\text{inv,SMR}} = c_{\text{inv,SMR}} \cdot IC_{\text{SMR}} \qquad (39)$$

$$C_{\text{om,SMR}} = \sum_{n=1}^{N} \frac{c_{\text{om,SMR}} \cdot IC_{\text{SMR}}}{(1+r)^n} \qquad (40)$$

$$C_{\text{gas,SMR}} = \sum_{n=1}^{N} \frac{\sum_{t=1}^{t=h}(c_{\text{gas}} \cdot G_{\text{gas}} \cdot H_{\text{SMR}}(t))}{(1+r)^n} \qquad (41)$$

$$C_{\text{grid,SMR}} = \sum_{n=1}^{N} \frac{\sum_{t=1}^{t=h}(c_{\text{grid}} \cdot EP_{\text{grid}} \cdot H_{\text{SMR}}(t))}{(1+r)^n} \qquad (42)$$

$$C_{\text{c,SMR}} = c_c \cdot LCCE_{\text{SMR}} \qquad (43)$$

$$0 \leq H_{\text{SMR}}(t) \leq IC_{\text{SMR}} \qquad (44)$$

$$0 \leq IC_{\text{SMR}} \qquad (45)$$

$$H_{\text{SMR}}(t) = \frac{H_{\text{output}}}{h} \qquad (46)$$

where $C_{\text{inv,SMR}}$, $C_{\text{om,SMR}}$, $C_{\text{gas,SMR}}$, $C_{\text{grid,SMR}}$, and $C_{\text{c,SMR}}$ are the investment cost, operation and maintenance cost, purchase cost of natural gas, power purchase cost and carbon emission cost of SMR, respectively. $c_{\text{inv,SMR}}$ is the investment cost of the reformer, \$/kW $H_2$. $IC_{\text{SMR}}$ is the capacity of the reformer, kW $H_2$. $c_{\text{om,SMR}}$ is the operation and maintenance cost of the reformer, \$/kW $H_2$-year. $G_{\text{gas}}$ is the natural gas required to produce 1 kg of hydrogen, kg. $EP_{\text{grid}}$ is electricity consumed to produce 1 kg of hydrogen, kWh. $LCCE_{\text{SMR}}$ is the LCCE of SMR, kg. $H_{\text{SMR}}$ is the hourly production of hydrogen of SMR, kg. Difference exists on investment costs, operation and maintenance costs, power consumption, natural gas consumption and direct coefficients of carbon emission between SMR + CCUS and SMR. The technical and economic parameters of the two systems are shown in Supplementary Table 4. $c_{\text{gas}}$ is the price of natural gas, \$/m³. The price of natural gas in each province is shown in Supplementary Table 5[47].

## Hydrogen production from industry by-products

The LCOH and LCCE of various hydrogen production technologies from industry by-product are shown in Supplementary Table 6.

## Environmental assessment

The life cycle carbon emissions of grid electricity and photovoltaic panels are considered in the water electrolysis hydrogen production system. The LCCE of the system can be calculated as follows.

$$LCCE_{WE} = \lambda_{PV} \cdot IC_{PV} + \sum_{n=1}^{N} \frac{\lambda_{grid} \cdot \sum_{t=1}^{t=h} EP_{grid}(t)}{(1+r)^n} \quad (47)$$

where $\lambda_{PV}$ is carbon emission factor throughout full life cycle of photovoltaic panels, 0.04 kg $CO_2$/kWh. $\lambda_{grid}$ is the grid carbon emission factor, the carbon emission factors of each province are shown in Supplementary Table 7. Grid carbon emissions are not included in the off-grid hydrogen production system.

The LCCE of SMR include direct carbon emissions, indirect carbon emissions generated by power purchase, carbon emissions associated with natural gas production and processing, and carbon emissions leaked during the natural gas production process[29]. The calculation is as follows.

$$LCCE_{SMR}$$
$$= \sum_{n=1}^{N} \frac{\sum_{t=1}^{t=h} CA_{SMR}(t) + \lambda_{grid} \cdot \sum_{t=1}^{t=h} EP_{grid}(t) + (\lambda_{gas} + \beta \cdot \delta) \cdot \sum_{t=1}^{t=h} G_{gas}(t)}{(1+r)^n} \quad (48)$$

where $CA_{SMR}$ is the direct carbon emissions of SMR, kg $CO_2$/kg $H_2$; $\lambda_{gas}$ is the carbon emission factor associated with natural gas production and processing, 0.3 kg $CO_2$/kg $CH_4$. $\beta$ is the fraction of the natural gas input that leaks during natural gas production and processing (kg $CH_4$ leaked/kg $CH_4$ input), 4%. $\delta$ is global warming potential of $CH_4$ relative to $CO_2$, the 20-year global warming potential of 85 kg $CO_2$/kg $CH_4$ is selected because of the urgency to decarbonize in the next few decades and the short-lived nature of natural gas in the atmosphere[29,48].

## Subsidy policies

The subsidized policies of hydrogen production issued by various provinces in China are researched and divided into three categories. The first category is electricity price concession, such as the Policy Measures to Optimize Energy Structure and Promote Green and Low Carbon Development in Chengdu Municipality[49]. The second category is investment matching incentives for carbon reduction, such as the Several Policies and Measures to Support the High-quality Development of Hydrogen Energy and Hydrogen Fuel Cell Vehicles issued by Shenyang[50]. The third category is production subsidies, such as the Relevant Support Policies of Karamay Municipality to Support the Development of Hydrogen Energy Industry[51]. Accordingly, three subsidy policies scenarios are set in this research.

## Reporting summary

Further information on research design is available in the Nature Portfolio Reporting Summary linked to this article.

## Data availability

The photovoltaic power generation data are available from the PVWatts Calculator of the National Renewable Energy Laboratory (https://pvwatts.nrel.gov/). All data to generate Figs. 2–8 are provided in the Source Data file. All other data are available in the Manuscript and the Supplementary Information. Source data are provided with this paper.

## Code availability

The code used in this study is available from the authors upon request.

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

## Acknowledgements

This research has been supported by National Key R&D Program of China (grant number: 2022YFB4004400, F.P. and B.S.), Major Program of National Natural Science Foundation of China (grant number: 62192753, B.S.), and the Science Fund for Creative Research Groups of the National Natural Science Foundation of China (grant number: 61821004, B.S.). We acknowledge Tianguang Lü, Guangsheng Pan, Xueliang Yuan, and Binbin Yu for their helpful comments and suggestions, as well as Xiaoming Xin, Wenchuan Zhang, and Tianyu Zhao for their contributions to the data collection.

## Author contributions

G.F. conceived and designed the work, performed the data collection and analysis, constructed the optimization model, and wrote the paper. H.Z. contributed to data validation and participated in writing the paper. B.S. and F.P. supervised this work and contributed to verification, review and editing the manuscript. All authors contributed to the discussion of the results and approved the manuscript.

## Competing interests

The authors declare no competing interests.
