## [Transparent Peer Review file · Nature Communications]

Economic and environmental competitiveness of multiple hydrogen production pathways in China

Corresponding Author: Professor Bo Sun

Version 0:

Reviewer comments:

Reviewer #1

(Remarks to the Author)

Dear author,

Rolling out a future hydrogen system is important to achieve greenhouse gas neutrality by the middle of this century. China is an important player for the future development due to its size and its economic strength.

Optimizations of hydrogen production for countries is not new. It is also not new for China, e.g. <https://doi.org/10.1016/B978-0-12-823377-1.50019-7>, <https://doi.org/10.1016/j.energy.2021.121193>. Thus, I do not see the novelty of your approach.

Maybe your article would be better suited in a different journal as a perspective paper focusing on the implications of your results.

Here some minor comments regarding your paper:

l. 27: hydrogen is not per se green and low-carbon it is a secondary energy carrier and depends on the primary energy source used for its production, which is up to today mainly fossil fuels.

l. 27f: Today hydrogen is mainly used in traditional industrial applications like ammonia production or in refineries. The sources 1 and 2 see probably wider applications in the future.

l. 28: energy is not consumed, it can only be transferred from one stage like wind energy into another like electricity.

l. 30: What is the global hydrogen production today?

l. 57f.: '... production via electrolysis using low-emission electricity ...' is not resulting in a '... carbon emission intensity of 24 kg CO₂/kg H₂ ...' That sound more like an electricity mix relying heavily on coal.

l. 53-62: please cite here the original cost results published by the papers and studies in €, \$ etc and not in ¥. You can give values in ¥ in brackets together with the source of currency exchange rate.

l. 70: were compared, not was compared

l. 126. '... In addition, the coal gasification (CG) and the CG combined with combined with carbon capture, utilization and storage (CG+CCUS) are ...'

l. 621: According to the IPCC methane has a global warming potential of ca. 28 kgCO₂/kgCH₄. Please check your source. Did you consider lower full load hours of the off-grid system compared to WE1-3?

(Remarks on code availability)

Reviewer #2

(Remarks to the Author)

The manuscript explores for four different H₂ production system configurations for the different provinces in China using MILP optimisation to determine LCOH and LCCE and considers policy impact and projected outcomes to 2050. The manuscript is generally clearly written, and the results interesting. I have suggestions for various improvements below.

1. Benefit to a broad readership maybe limited. Firstly, the currency is in Yuan relevant for the China provinces as the specific case study. Secondly the lines 110-113 highlight the readership interest area: "The configuration optimization results can provide guidance for investors in hydrogen production projects, the evaluation results of hydrogen production

system in different provinces [in China] can provide reference for national and provincial energy policy makers." To broaden the readership, it is suggested that the outcomes from this study are used to provide insights to how the global hydrogen industry can be structured and developed to achieve low LCOH and LCCE. That is, what are the best policies, system configurations and grid characteristics that achieve best outcomes?

2. A clearer statement of novelty is needed. Novelty appears limited to an analysis that covers many provinces in China and the outcome that locations with low C intensity grids are more beneficial for H₂ production via grid connection appears obvious. The different configurations are not novel and have been assessed by others in different case studies. Is there also novelty in the use and application of MILP optimisation and/or the inclusion of specific policies not reported before?

3. The study of how policy impacts the outcomes will be of interest for many countries. Of relevance are lines 328-331 "This suggests that different policies are applicable to different hydrogen production systems via water electrolysis, targeted policies should be issued." A clear summary of policy impact to LCOH and LCCE will be of interest.

4. Figure 4 result: Why does province BJ have large H₂ storage but low compressor rating in Fig 4b for WE3?

5. General - Figure's -the readability (Font size) needs improving.

6. The manuscript will benefit from adding a conclusion.

7. Method section

a. Why is wind not included?

b. Why is there a constraint of using hydrogen production rate at only 1kg/hr or 8760kg/year? Is this normalisation? Does the outcome change when economies of scale are considered?

c. Lines 399-401. WE1. The capacity of the electrolyser is the same for each province so what is being optimised?

d. Line 408. The statement "...there is no need for energy storage" needs explaining. Why are WE1 and WE2 configurations having no compression or storage? This implies an assumption that offtake use applications uses H₂ at the same rate of H₂ production and at the low hydrogen pressure provided from the electrolyser. Is this assumption valid in all provinces?

e. Line 392-393 "this study did not consider the actual hydrogen demand, which is different in each province." However, is it true that the Western provinces are further from large population centres and may have a greater cost in storage and distribution to the eastern provinces that changes the LCOH and LCCE. The assumption in this work that off-take dynamics and quantity is the same in all provinces may need investigating in this manuscript rather than for future work.

f. Line 410 WE3. "... renewable energy accounts for 50% of the total hydrogen production." Why is this constraint used rather than RE percentage from solar being a variable determined by optimisation?

g. Line 438 "The reliability of hydrogen is ensured at a constant production rate per hour, which is necessary for hydrogen plants". This implies an assumption that there is little/no degradation of electrolyser efficiency when operated at constant power. Is there also a case where the electrolyser technology can today or next generations have little/no degradation when power cycling and if so, what is the outcome when the electrolyser is operated dynamically? Or, what is the result when operating an electrolyser dynamically with a larger degradation rate compared to an electrolyser operating in constant power condition?

(Remarks on code availability)

Version 1:

Reviewer comments:

Reviewer #1

(Remarks to the Author)

Dear authors,

you improved your article significantly. Congratulations!

(Remarks on code availability)

Reviewer #2

(Remarks to the Author)

The revised manuscript is a noteworthy and original analysis of for four different H₂ production system configurations for the different provinces in China using MILP optimisation to determine LCOH and LCCE and considers policy impact and projected outcomes to 2050. The method and analysis appears sound, and the results well described and are at expected standards. I thank the authors for their answers to my review questions and editing the manuscript. Revisions made have improved the manuscript and I recommend the work is published.

(Remarks on code availability)

REVIEWER COMMENTS

Reviewer #1 (Remarks to the Author):

Dear author,

Rolling out a future hydrogen system is important to achieve greenhouse gas neutrality by the middle of this century. China is an important player for the future development due to its size and its economic strength.

Optimizations of hydrogen production for countries is not new. It is also not new for China, e.g. <https://doi.org/10.1016/B978-0-12-823377-1.50019-7>, <https://doi.org/10.1016/j.energy.2021.121193>. Thus, I do not see the novelty of your approach. Maybe your article would be better suited in a different journal as a perspective paper focusing on the implications of your results.

Here some minor comments regarding your paper:

l. 27: hydrogen is not per se green and low-carbon it is a secondary energy carrier and depends on the primary energy source used for its production, which is up to today mainly fossil fuels.

l. 27f: Today hydrogen is mainly used in traditional industrial applications like ammonia production or in refineries. The sources 1 and 2 see probably wider applications in the future.

l. 28: energy is not consumed, it can only be transferred from one stage like wind energy into another like electricity.

l. 30: What is the global hydrogen production today?

l. 57f.: ‘... production via electrolysis using low-emission electricity ...’ is not resulting in a ‘... carbon emission intensity of 24 kg CO₂/kg H₂ ...’ That sound more like an electricity mix relying heavily on coal.

l. 53-62: please cite here the original cost results published by the papers and studies in €, \$ etc and not in ¥. You can give values in ¥ in brackets together with the source of currency exchange rate.

l. 70: were compared, not was compared

l. 126. ‘... In addition, the coal gasification (CG) and the CG combined with combined with carbon capture, utilization and storage (CG+CCUS) are ...’

l. 621: According to the IPCC methane has a global warming potential of ca. 28 kgCO₂/kgCH₄. Please check your source.

Did you consider lower full load hours of the off-grid system compared to WE1-3?

Reviewer #2 (Remarks to the Author):

The manuscript explores for four different H₂ production system configurations for the different provinces in China using MILP optimisation to determine LCOH and LCCE and considers policy impact and projected outcomes to 2050. The manuscript is generally clearly written, and the results interesting. I have suggestions for various improvements below.

1. Benefit to a broad readership maybe limited. Firstly, the currency is in Yuan relevant for the China provinces as the specific case study. Secondly the lines 110-113 highlight the readership interest area: “The configuration optimization results can provide guidance for investors in hydrogen production projects, the evaluation results of hydrogen production system in different provinces [in China] can provide reference for national and provincial energy policy makers.” To broaden the readership, it is suggested that the outcomes from this study are used to provide insights to how the global hydrogen industry can be structured and developed to achieve low LCOH and LCCE. That is, what are the best policies, system configurations and grid characteristics that achieve best outcomes?

2. A clearer statement of novelty is needed. Novelty appears limited to an analysis that covers many provinces in China and the outcome that locations with low C intensity grids are more beneficial for H₂ production via grid connection appears obvious. The different configurations are not novel and have been assessed by others in different case studies. Is there also novelty in the use and application of MILP optimisation and/or the inclusion of specific policies not reported before?

3. The study of how policy impacts the outcomes will be of interest for many countries. Of relevance are lines 328-331 “This suggests that different policies are applicable to different hydrogen production systems via water electrolysis, targeted policies should be issued.” A clear summary of policy impact to LCOH and LCCE will be of interest.

4. Figure 4 result: Why does province BJ have large H₂ storage but low compressor rating in Fig 4b for WE3?

5. General - Figure's -the readability (Font size) needs improving.

6. The manuscript will benefit from adding a conclusion.

7. Method section

a. Why is wind not included?

b. Why is there a constraint of using hydrogen production rate at only 1kg/hr or 8760kg/year? Is this normalisation? Does the outcome change when economies of scale are considered?

c. Lines 399-401. WE1. The capacity of the electrolyser is the same for each province so what is being optimised?

d. Line 408. The statement “...there is no need for energy storage” needs explaining. Why are WE1 and WE2 configurations having no compression or storage? This implies an assumption that offtake use applications uses H₂ at the same rate of H₂ production and at the low hydrogen pressure provided from the electrolyser. Is this assumption

valid in all provinces?

e. Line 392-393 “this study did not consider the actual hydrogen demand, which is different in each province.” However, is it true that the Western provinces are further from large population centres and may have a greater cost in storage and distribution to the eastern provinces that changes the LCOH and LCCE. The assumption in this work that off-take dynamics and quantity is the same in all provinces may need investigating in this manuscript rather than for future work.

f. Line 410 WE3. “... renewable energy accounts for 50% of the total hydrogen production.” Why is this constraint used rather than RE percentage from solar being a variable determined by optimisation?

g. Line 438 “The reliability of hydrogen is ensured at a constant production rate per hour, which is necessary for hydrogen plants”. This implies an assumption that there is little/no degradation of electrolyser efficiency when operated at constant power. Is there also a case where the electrolyser technology can today or next generations have little/no degradation when power cycling and if so, what is the outcome when the electrolyser is operated dynamically? Or, what is the result when operating an electrolyser dynamically with a larger degradation rate compared to an electrolyser operating in constant power condition?

Point-by-point response to the reviewers' comments

Manuscript Number: NCOMMS-24-38600A

Title: Economic and environmental competitiveness of multiple hydrogen production pathways in China

To Editor and Reviewers:

I, on behalf all authors of this paper, express our great appreciation to your valuable work to help us review this paper. According to your useful comments and suggestions, we have revised this paper point-by-point, which are highlighted in the revised manuscript.

In what follows, we detail the revisions made in response to the reviewers' suggestions. For convenience, the reviewers' comments and suggestions are marked in black text, our responses in **red text**, and the revisions in the manuscript in **blue text**.

Many thanks go to your valuable help.

Best wishes,

Bo Sun

Response to Editorial Requests

Reply: In addition to the responses to the reviewers and the corresponding revisions in the manuscript and supplementary material, we have also revised the format of the manuscript following the formatting instructions.

Response to the reviewer's comments:

Reviewer #1:

Dear author,

Rolling out a future hydrogen system is important to achieve greenhouse gas neutrality by the middle of this century. China is an important player for the future development due to its size and its economic strength.

Reply: We appreciate your positive response and recognition of the research in this paper. The manuscript has been carefully revised according to the comments.

Optimizations of hydrogen production for countries is not new. It is also not new for China, e.g. <https://doi.org/10.1016/B978-0-12-823377-1.50019-7>, <https://doi.org/10.1016/j.energy.2021.121193>. Thus, I do not see the novelty of your approach. Maybe your article would be better suited in a different journal as a

perspective paper focusing on the implications of your results.

Reply: Thank you very much for your advice. This suggestion has important guiding significance for our thesis writing and research work. The novelty of this paper are as follows:

In terms of the research object, the above two literatures and other papers on the optimization of hydrogen production in China were conducted only on a national spatial scale for a particular type of hydrogen production system *via* water electrolysis, without considering the differences in system configurations and provincial resources. To fill the gap, this paper optimizes and evaluates hydrogen production system *via* water electrolysis with different configurations in all provinces of the China. It is significant to conduct the study for the following reasons. Firstly, electricity for hydrogen production *via* water electrolysis may be derived from the grid or renewable energy sources, leading to different configurations and cost. Secondly, there are significant differences in renewable resources, energy prices, and grid facilities among provinces of China. Finally, provincial hydrogen energy plans and development policies, such as *Mid-to-Long-Term Plan for Hydrogen Energy of Shandong Province*, *Implementation Plan for Development of the Hydrogen Energy Industry of Beijing (2021-2025)*, are reviewed and approved by the China National Energy Administration based on the resource endowment, energy development capacity, and environmental carrying capacity of each province.

In terms of methods, the above two literatures established monthly and yearly supply-demand balance hydrogen energy optimization models, respectively. To guarantee the hourly reliability of the hydrogen supply, a model for configuration and full-year hourly operation optimization of hydrogen production systems is developed in this paper. The system investment and operation schemes in different provinces can be jointly optimized under different subsidy policies for different target years.

Finally, the impact of policy on the development of multiple hydrogen production pathways in China remains unclear. Therefore, the development of multiple hydrogen production pathways (hydrogen production *via* water electrolysis, coal, natural gas, and industry by-products) in the future scenario from 2025 to 2050 is discussed. The changes of the development pathways under three actual subsidy policies, *i.e.*, electricity price concession, investment matching incentives for carbon reduction, production subsidies, are analyzed.

The research gaps and innovations are clarified in the new version as follows. And the optimization method is described in detail.

(Line 69 to Line 116, Introduction)

With the breakthrough of renewable energy and the technology of electrolyzer in China, many scholars have begun to explore the economic and environmental feasibility of hydrogen production *via* water electrolysis in China. The cost and carbon emissions of various hydrogen production pathways in China were compared by IEA firstly, including hydrogen production *via* grid electricity, renewable energy electricity, natural gas and coal¹⁷. Yang et al.¹⁸ investigated the potential role of clean hydrogen

by focusing on the industries facing bottlenecks in carbon reduction in China, i.e., heavy industry and heavy transportation. The results showed that clean hydrogen can significantly reduce carbon emissions from heavy industry. And about 1.72 trillion dollars in new investments can be avoided in clean hydrogen scenario that reaches 65.7 million tons of production in 2060. Song et al.¹⁹ explored the feasibility of transporting hydrogen in liquid form to Japan, which produced by offshore wind power electrolysis in China. The results showed that China's offshore energy deliveries and costs are in line with Japan's idealized future projections. Pan et al.²⁰ provided a detailed assessment of the cost of hydrogen production system combined photovoltaic and grid in China, using a fixed daily supply of hydrogen as a requirement. Li et al.²¹ compared the average cost of hydrogen production *via* electricity and coal in China using a hydrogen optimization model with monthly supply and demand balances. Qiu et al.²² calculated the total hydrogen demand in China from 2017 to 2060 using an annual supply and demand balance energy planning model.

However, the above researches were conducted only on a national spatial scale for a particular type of hydrogen production system *via* water electrolysis, without considering the differences in system configurations and provincial resources. Firstly, electricity for hydrogen production *via* water electrolysis may be derived from the grid or renewable energy sources, leading to different configurations and cost. Secondly, there are significant differences in renewable resources, energy prices, and grid facilities among provinces of China. Finally, provincial hydrogen energy plans and development policies, such as *Mid-to-Long-Term Plan for Hydrogen Energy of Shandong Province*²³, *Implementation Plan for Development of the Hydrogen Energy Industry of Beijing (2021-2025)*²⁴, are reviewed and approved by the China National Energy Administration based on the resource endowment, energy development capacity, and environmental carrying capacity of each province. Therefore, it is of great significance to evaluate hydrogen production system *via* water electrolysis with different configurations in all provinces of the China in this context.

Here, a model for configuration and full-year hourly operation optimization of hydrogen production systems is developed considering the hourly hydrogen supply reliability. The data are derived from real-world electricity prices and grid carbon emission factors for 31 provinces, as well as hourly photovoltaic power generation from the National Renewable Energy Laboratory. The system investment and operation schemes in different provinces are jointly optimized under different subsidy policies for different target years. Then, the levelized cost of hydrogen (LCOH) and life cycle carbon emissions (LCCE) of different hydrogen production systems *via* water electrolysis are quantified for 31 provinces in China, including grid, photovoltaic and grid combined, fixed renewable energy penetration rate, and off-grid. The development of multiple hydrogen production pathways in the future scenario from 2025 to 2050 is discussed. The changes of the development pathways under three actual subsidy policies, *i.e.*, electricity price concession, investment matching incentives for carbon reduction, production subsidies, are analyzed.

21. Li T, Liu P, Li Z. Modelling and Optimization of a Multi-regional Hydrogen Supply System: a Case Study of China. In: *30th European Symposium on Computer Aided Process Engineering* (2020).
22. Qiu S, Lei T, Wu J, Bi S. Energy demand and supply planning of China through 2060. *Energy* **234**, (2021).

(Line 554 to Line 575, Methods)

Hydrogen production system optimization model

The hourly power generation data of photovoltaic panels with a rated power of 1 kW are collected from the PVWatts Calculator developed by the National Renewable Energy Laboratory, USA ⁴¹. The electricity price comes from the electricity price sales table of each provincial power grid company. The above data and the carbon emission factors of each provincial power grid and the technical-economic parameters of the hydrogen production system equipment are inputted into the hydrogen production system optimization model, and the model framework is presented in Figure 9. An optimization model for the configuration and full-year hourly operation of the hydrogen production system *via* water electrolysis is established, considering multiple constraints such as equipment operation, power purchases, electricity balance, hydrogen balance, and hydrogen supply reliability. Through this model, the investment and operation schemes of multiple hydrogen production systems *via* water electrolysis in different provinces for China can be jointly optimized under different subsidy policies for different target years, while the LCCE of the systems can be assessed. The hydrogen production system optimization model is constructed as a mixed-integer linear programming (MILP) model. MILP, which efficiently solve linear equations and ensure the balance between computational efficiency and robustness, has become a predominant optimization method for the design and operation of energy systems and has been widely used. Based on the Matlab platform, the Gurobi solver is invoked through the Yalmip toolbox to solve the model.

Figure 9. Modeling framework for hydrogen production system optimization.

Here some minor comments regarding your paper:

l. 27: hydrogen is not per se green and low-carbon it is a secondary energy carrier and depends on the primary energy source used for its production, which is up to today mainly fossil fuels.

Reply: Sorry for the confusion. Hydrogen energy is not low-carbon per se, it is a secondary energy source. We have rephrased this point as follows.

(Line 27 to Line 28, Introduction)

Hydrogen is a secondary energy with abundant sources, with global production currently exceeding 95 million tons ¹.

l. 27f: Today hydrogen is mainly used in traditional industrial applications like ammonia production or in refineries. The sources 1 and 2 see probably wider applications in the future.

Reply: Thanks for your suggestion. Hydrogen is mainly used in traditional industries at present, and Literature 1 and 2 are an outlook for a wide range of future applications. This point is clarified as follows.

(Line 28 to Line 31, Introduction)

In the future hydrogen energy may achieve a wide range of applications ^{2, 3}, promoting a high share of renewable energy development and accelerating the decarbonization of industry, transportation, buildings and other sectors ^{4, 5}.

(Line 473 to Line 477, Methods)

Each hydrogen production pathway needs to achieve the same hydrogen supply reliability (hydrogen supply rate) for a fair comparison of LCOH. More than 90% of the global hydrogen used in industrial applications such as refining, ammonia synthesis, and methanol preparation, a stable supply of hydrogen is an absolute requirement for safe industrial production ¹².

l. 28: energy is not consumed, it can only be transferred from one stage like wind energy into another like electricity.

Reply: We fully agree with your comments. Energy is not consumed. Hydrogen can be produced from renewable energy, thus contributing to the development of renewable energy. This point is modified as follows.

(Line 28 to Line 32, Introduction)

In the future hydrogen energy may achieve a wide range of applications ^{2, 3}, promoting a high share of renewable energy development and accelerating the decarbonization of industry, transportation, buildings and other sectors ^{4, 5}. According to predictions, the demand for hydrogen energy will increase to 660 million tons by 2050, account for 22% of global terminal energy demand.

l. 30: What is the global hydrogen production today?

Reply: According to the International Energy Agency, global hydrogen production now exceeds 95 million tons. This is clarified in the Introduction.

(Line 27 to Line 28, Introduction)

Hydrogen is a secondary energy with abundant sources, with global production currently exceeding 95 million tons ¹.

1. *Global Hydrogen Review 2023*. International Energy Agency (2023).

l. 57f.: ‘... production *via* electrolysis using low-emission electricity ...’ is not resulting in a ‘... carbon emission intensity of 24 kg CO₂/kg H₂ ...’ That sound more like an electricity mix relying heavily on coal.

Reply: Sorry for the confusion. This is revised in the new version as follows.

(Line 58 to Line 61, Introduction)

The cost of hydrogen production *via* electrolysis exceeded 3.4 \$/kg H₂, with a carbon emission intensity of 24 kg CO₂/kg H₂ for using grid electricity, and close to zero for using renewable energy electricity without considering the manufacturing of photovoltaics or wind turbines.

l. 53-62: please cite here the original cost results published by the papers and studies in €, \$ etc and not in ¥. You can give values in ¥ in brackets together with the source of currency exchange rate.

Reply: Thank you very much for your advice. We use the original cost results in the new version, as shown below. In addition, combining with another reviewer's comment, the currency units in this paper are modified to \$ for the purpose of broadening the readership.

(Line 58 to Line 65, Introduction)

The cost of hydrogen production *via* electrolysis exceeded 3.4 \$/kg H₂, with a carbon emission intensity of 24 kg CO₂/kg H₂ for using grid electricity, and close to zero for using renewable energy electricity without considering the manufacturing of photovoltaics or wind turbines. Guerra et al. ¹⁴ focused on 20 states in the United States, simulated the operation of electrolyzer dynamically and demonstrated that electrolysis equipment can provide hydrogen with cost competitiveness. Glenk et al. ¹⁵ pointed that the cost of hydrogen production *via* renewable energy in Germany is 3.23 €/kg, already achieving cost competitiveness.

l. 70: were compared, not was compared

Reply: Thanks for your careful suggestion. This is revised in the new version.

(Line 71 to Line 74, Introduction)

The cost and carbon emissions of various hydrogen production pathways in China were compared by IEA firstly, including hydrogen production *via* grid electricity, renewable energy electricity, natural gas and coal ¹⁷.

l. 126. ‘... In addition, the coal gasification (CG) and the CG combined with combined with carbon capture, utilization and storage (CG+CCUS) are ...’

Reply: Thank you very much for your careful review. The error is revised in the new

version and is checked in the whole article.

(Line 159 to Line 163, Results)

In addition, the coal gasification (CG) and the CG combined with carbon capture, utilization and storage (CG+CCUS) are selected to represent hydrogen production *via* coal, steam methane reformer (SMR) and SMR combined with carbon capture, utilization and storage (SMR+CCUS) are selected to represent hydrogen production *via* natural gas.

l. 621: According to the IPCC methane has a global warming potential of ca. 28 kgCO₂/kgCH₄. Please check your source.

Reply: Thank you very much for your advice. All of your suggestions are very important.

According to the U.S. Environmental Protection Agency¹, “Understanding Global Warming Potentials,” (2022). <https://www.epa.gov/ghgemissions/understanding-global-warming-potentials>. Methane (CH₄) is estimated to have a global warming potential (GWP) of 27-30 over 100 years. The 20-year GWP is sometimes used as an alternative to the 100-year GWP. Just like the 100-year GWP is based on the energy absorbed by a gas over 100 years, the 20-year GWP is based on the energy absorbed over 20 years. This 20-year GWP prioritizes gases with shorter lifetimes, because it does not consider impacts that happen more than 20 years after the emissions occur. For example, for CH₄, which has a short lifetime, the 100-year GWP is much less than the 20-year GWP. We choose the 20-year GWP (85 kg CO₂/kg CH₄) because of the urgency to decarbonize in the next few decades and the short-lived nature of natural gas in the atmosphere².

1. *Understanding Global Warming Potentials*. U.S. Environmental Protection Agency. <https://www.epa.gov/ghgemissions/understanding-global-warming-potentials> (2022).
2. Bracci JM, Sherwin ED, Boness NL, Brandt AR. A cost comparison of various hourly-reliable and net-zero hydrogen production pathways in the United States. *Nature Communications* **14**, (2023).

This is clearly clarified in the new version.

(Line 720 to Line 723, Methods)

δ is global warming potential of CH₄ relative to CO₂, the 20-year global warming potential of 85 kg CO₂/kg CH₄ is selected because of the urgency to decarbonize in the next few decades and the short-lived nature of natural gas in the atmosphere^{29, 48}.

29. Bracci JM, Sherwin ED, Boness NL, Brandt AR. A cost comparison of various hourly-reliable and net-zero hydrogen production pathways in the United States. *Nature Communications* **14**, (2023).
48. *Understanding Global Warming Potentials*. U.S. Environmental Protection Agency. <https://www.epa.gov/ghgemissions/understanding-global-warming-potentials> (2022).

Did you consider lower full load hours of the off-grid system compared to WE1-3?

Reply: Thank you very much for your advice. This suggestion plays a significant role in im-proving the quality of this article.

It is certain that compared to WE1 and WE2, the full load operation hours of the electrolyzer in WE3 and WE4 are substantially lower. In the case of Beijing, the annual full load operation hours in WE3 and WE4 are only 1679 h and 1107 h respectively. WE4 are operated intermittently. The degradation rates of the four systems are 2.5%, 2.5%, 2.2%, and 1.9%, respectively, with only a small impact on the lifetime.

This is supplemented in the new version.

(Line 247 to Line 258, Results)

In addition, compared to WE1 and WE2, the full load operation hours of the electrolyzer in WE3 and WE4 are substantially lower. In the case of Beijing, the annual full load operation hours in WE3 and WE4 are only 1679 h and 1107 h respectively, and the full load operation rates are 19.2% and 12.6%, the degradation rates are 2.2% and 1.9%, with only a small impact on the lifetime, as shown in Section S3 of the Supplementary materials. The degradation rates for constant and dynamic power operation at the same capacity are both 1.9%, using WE4 as an example. If the alkaline (ALK) electrolyzer is replaced, the degradation rates for the same operation condition will be 14.3% and 15.9%. The dynamic power operation has less effect on proton exchange membrane (PEM) electrolyzer but increases the degradation rate of ALK electrolyzer. And the rationale of adopting the PEM electrolyzer is validated.

(Supplementary materials)

S3. Operation characteristics of hydrogen production system *via* water electrolysis in Beijing

Figure S2. Hourly electric power of electrolyzer in a typical month.

(a) WE3; (b) WE4.

The degradation rate of the electrolyzer (D_{EL}) is calculated as follows ¹⁵.

$$D_{EL} = \frac{H_{sto,EL} + \frac{1}{h} \times \sum_{t=1}^{t=h} \left(\frac{EP_{EL}(t)}{IC_{EL}} \times (\sigma_{EL} - H_{sto,EL}) + k_{EL} f_{EL}(t)^2 \right)}{N_{EL}} \quad (S41)$$

$$f_{EL}(t) = \frac{|EP_{EL}(t) - EP_{EL}(t-1)|}{IC_{EL}} \quad (S42)$$

where, EP_{EL} is the electric power of the electrolyzer, kW. N_{EL} is the lifetime of the electrolyzer, year. $H_{sto,EL}$, σ_{EL} , k_{EL} are empirical coefficients related to the degradation rate, the proton exchange membrane (PEM) electrolyzer is taken as 0.35, 0.5, 1×10^{-5} , and the alkaline (ALK) electrolyzer is taken as 2, 10, 0.1, respectively. f_{EL} is the power volatility of the electrolyzer.

Table S8. Full load operation time and degradation rate of four hydrogen production systems *via* water electrolysis

System	WE1	WE2	WE3	WE4
Full load operation hours (h)	8760	8760	1679	1107
Full load operation rate (%)	100	100	19.2	12.6
Degradation rate (%)	2.5	2.5	2.2	1.9

Table S9. Comparison of electrolyzer degradation rates between constant power operation and dynamic operation

System	Constant power operation	Dynamic operation
PEM electrolyzer (%)	1.9	1.9
ALK electrolyzer (%)	14.3	15.9

(Taking WE4 as an example, the hydrogen production rate of the electrolyzer is 1kg/h in constant power operation.)

- Yu B, Fan G, Sun K, Chen J, Sun B, Tian P. Adaptive energy optimization strategy of island renewable power-to-hydrogen system with hybrid electrolyzers structure. *Energy* **301**, (2024).

Reviewer #2:

The manuscript explores for four different H₂ production system configurations for the different provinces in China using MILP optimisation to determine LCOH and LCCE and considers policy impact and projected outcomes to 2050. The manuscript is generally clearly written, and the results interesting. I have suggestions for various improvements below.

Reply: We appreciate the Reviewer's invaluable suggestions and comments. Your positive affirmation of our work prompted us to further improve the quality of this work. The manuscript has been carefully revised according to the Reviewer's comments.

1. Benefit to a broad readership maybe limited. Firstly, the currency is in Yuan relevant for the China provinces as the specific case study. Secondly the lines 110-113 highlight the readership interest area: "The configuration optimization results can provide guidance for investors in hydrogen production projects, the evaluation results of hydrogen production system in different provinces [in China] can provide reference for national and provincial energy policy makers." To broaden the readership, it is suggested that the outcomes from this study are used to provide insights to how the global hydrogen industry can be structured and developed to achieve low LCOH and LCCE. That is, what are the best policies, system configurations and grid characteristics that achieve best outcomes?

Reply: Thank you very much for your advice. This suggestion has important guiding significance for our thesis writing and research work.

For the benefit of a broader readership, firstly, the currency of this paper is revised to US dollars throughout, with examples as follows. Secondly, the insights provided for the development of the global hydrogen production industry are supplemented in the new version.

(Line 170 to Line 174, Results)

The LCOH and LCCE of multiple hydrogen production pathways in 31 provinces of China is shown in Figure 2. From the perspective of different provinces, there are significant difference in the LCOH and LCCE of hydrogen production *via* water electrolysis across 31 provinces. The LCOH of the WE1 system is 4.6 \$/kg H₂-7.9 \$/kg H₂, and the LCOH of the WE2 system is 4.6 \$/kg H₂-7.6 \$/kg H₂.

(Line 138 to Line 146, Introduction)

Here, we show two important insights on the development of the global hydrogen production industry: (1) Grid-connected hydrogen production *via* water electrolysis has economic advantages only in areas where renewable energy is abundant or electricity prices are low, where electricity price concessions and production subsidy policies can reduce LCOH. (2) Off-grid hydrogen production system *via* water electrolysis may be the most economical and low-carbon pathway in the future, with carbon reduction incentives and production subsidy policies accelerating the realization. Before it becomes economically advantageous, hydrogen production *via* industrial by-products

is a good alternative.

2. A clearer statement of novelty is needed. Novelty appears limited to an analysis that covers many provinces in China and the outcome that locations with low C intensity grids are more beneficial for H₂ production *via* grid connection appears obvious. The different configurations are not novel and have been assessed by others in different case studies. Is there also novelty in the use and application of MILP optimisation and/or the inclusion of specific policies not reported before?

Reply: Thank you very much for your advice. All of your suggestions are very important. They have important guiding significance for our thesis writing and research work. The novelty of this paper are as follows:

In terms of the research object, the previous studies on the optimization of hydrogen production in China were conducted only on a national spatial scale for a particular type of hydrogen production system *via* water electrolysis, without considering the differences in system configurations and provincial resources. To fill the gap, this paper optimizes and evaluates hydrogen production system *via* water electrolysis with different configurations in all provinces of the China. It is significant to conduct the study for the following reasons. Firstly, electricity for hydrogen production *via* water electrolysis may be derived from the grid or renewable energy sources, leading to different configurations and cost. Secondly, there are significant differences in renewable resources, energy prices, and grid facilities among provinces of China. Finally, provincial hydrogen energy plans and development policies, such as *Mid-to-Long-Term Plan for Hydrogen Energy of Shandong Province*, *Implementation Plan for Development of the Hydrogen Energy Industry of Beijing (2021-2025)*, are reviewed and approved by the China National Energy Administration based on the resource endowment, energy development capacity, and environmental carrying capacity of each province.

In terms of methods, the previous studies only considered the daily, monthly, or yearly reliability of the hydrogen supply to establish the hydrogen production system optimization model. Reliable hourly supply of hydrogen is necessary for hydrogen users. Therefore, to guarantee the hourly reliability, a model for configuration and full-year hourly operation optimization of hydrogen production systems is developed. The system investment and operation schemes in different provinces are jointly optimized under different subsidy policies for different target years.

Finally, the impact of policy on the development of multiple hydrogen production pathways in China remains unclear. Therefore, the development of multiple hydrogen production pathways (hydrogen production *via* water electrolysis, coal, natural gas, and industry by-products) in the future scenario from 2025 to 2050 is discussed. The changes of the development pathways under three actual subsidy policies, *i.e.*, electricity price concession, investment matching incentives for carbon reduction, production subsidies, are analyzed.

The novelty is clearly stated in the Introduction section of the new version, and the

optimization model is detailed in the Methods section, as follows.

(Line 89 to Line 116, Introduction)

However, the above researches were conducted only on a national spatial scale for a particular type of hydrogen production system *via* water electrolysis, without considering the differences in system configurations and provincial resources. Firstly, electricity for hydrogen production *via* water electrolysis may be derived from the grid or renewable energy sources, leading to different configurations and cost. Secondly, there are significant differences in renewable resources, energy prices, and grid facilities among provinces of China. Finally, provincial hydrogen energy plans and development policies, such as *Mid-to-Long-Term Plan for Hydrogen Energy of Shandong Province*²³, *Implementation Plan for Development of the Hydrogen Energy Industry of Beijing (2021-2025)*²⁴, are reviewed and approved by the China National Energy Administration based on the resource endowment, energy development capacity, and environmental carrying capacity of each province. Therefore, it is of great significance to evaluate hydrogen production system *via* water electrolysis with different configurations in all provinces of the China in this context.

Here, a model for configuration and full-year hourly operation optimization of hydrogen production systems is developed considering the hourly hydrogen supply reliability. The data are derived from real-world electricity prices and grid carbon emission factors for 31 provinces, as well as hourly photovoltaic power generation from the National Renewable Energy Laboratory. The system investment and operation schemes in different provinces are jointly optimized under different subsidy policies for different target years. Then, the levelized cost of hydrogen (LCOH) and life cycle carbon emissions (LCCE) of different hydrogen production systems *via* water electrolysis are quantified for 31 provinces in China, including grid, photovoltaic and grid combined, fixed renewable energy penetration rate, and off-grid. The development of multiple hydrogen production pathways in the future scenario from 2025 to 2050 is discussed. The changes of the development pathways under three actual subsidy policies, *i.e.*, electricity price concession, investment matching incentives for carbon reduction, production subsidies, are analyzed.

(Line 554 to Line 575, Methods)

Hydrogen production system optimization model

The hourly power generation data of photovoltaic panels with a rated power of 1 kW are collected from the PVWatts Calculator developed by the National Renewable Energy Laboratory, USA⁴¹. The electricity price comes from the electricity price sales table of each provincial power grid company. The above data and the carbon emission factors of each provincial power grid and the technical-economic parameters of the hydrogen production system equipment are inputted into the hydrogen production system optimization model, and the model framework is presented in Figure 9. An optimization model for the configuration and full-year hourly operation of the hydrogen production system *via* water electrolysis is established, considering multiple constraints such as equipment operation, power purchases, electricity balance, hydrogen balance,

and hydrogen supply reliability. Through this model, the investment and operation schemes of multiple hydrogen production systems *via* water electrolysis in different provinces for China can be jointly optimized under different subsidy policies for different target years, while the LCCE of the systems can be assessed. The hydrogen production system optimization model is constructed as a mixed-integer linear programming (MILP) model. MILP, which efficiently solve linear equations and ensure the balance between computational efficiency and robustness, has become a predominant optimization method for the design and operation of energy systems and has been widely used. Based on the Matlab platform, the Gurobi solver is invoked through the Yalmip toolbox to solve the model.

Figure 9. Modeling framework for hydrogen production system optimization.

3. The study of how policy impacts the outcomes will be of interest for many countries. Of relevance are lines 328-331 “This suggests that different policies are applicable to different hydrogen production systems *via* water electrolysis, targeted policies should be issued.” A clear summary of policy impact to LCOH and LCCE will be of interest.

Reply: Thank you very much for your advice. This suggestion plays a significant role in improving the quality of this article. The impacts of the policy are clearly summarized in the Introduction and Results sections, below.

(Line 138 to Line 146, Introduction)

Here, we show two important insights on the development of the global hydrogen production industry: (1) Grid-connected hydrogen production *via* water electrolysis has economic advantages only in areas where renewable energy is abundant or electricity prices are low, where electricity price concessions and production subsidy policies can reduce LCOH. (2) Off-grid hydrogen production system *via* water electrolysis may be the most economical and low-carbon pathway in the future, with carbon reduction incentives and production subsidy policies accelerating the realization. Before it becomes economically advantageous, hydrogen production *via* industrial by-products is a good alternative.

(Line 393 to Line 410, Results)

The tendency of the cost competition of hydrogen production *via* water electrolysis under the scenario 2, 3, and 4 is shown in Figure 8 (c), (d) and (e). The subsidy policy is conducive to driving down the LCOH of hydrogen production systems *via* water electrolysis and accelerating the progress towards the time when they are cost-competitive. Compared to the baseline scenario, the first key point will be advanced by the three subsidy policies, about 5 years, 5 years, and 15 years, respectively. Under carbon reduction incentives and production subsidies, the second key point will be advanced by about 5 years and 15 years, respectively, *i.e.*, the off-grid hydrogen production *via* water electrolysis will become the most economical and low-carbon hydrogen production pathway much earlier. In addition, the electricity price subsidy policy has no effect on WE4. Because the investment cost accounts for a small proportion, the investment incentive policy for reducing carbon has a small impact on WE1 and WE2. The production subsidy will simultaneously promote the reduction of the LCOH of four hydrogen production systems *via* water electrolysis. This suggests that different policies are applicable to different hydrogen production systems *via* water electrolysis, targeted policies should be issued and promoted by each province according to their natural conditions and the type of hydrogen production *via* water electrolysis which is suitable for development.

4. Figure 4 result: Why does province BJ have large H₂ storage but low compressor rating in Fig 4b for WE3?

Reply: Thank you very much for your advice. The capacity of the compressor is determined by the actual maximum import rate of the hydrogen storage tank. The variation of hydrogen quality in the hydrogen storage tank in a typical month of Beijing is shown in Figure R1, and the rate of hydrogen import is shown in Figure R2. Although the system needs to be equipped with a larger capacity hydrogen storage tank (19.1 kg) to regulate the hydrogen supply, the smaller capacity of the electrolyzer and the smaller amount of hydrogen produced hour by hour resulted in a maximum import rate of the tank of 0.87 kg/h in actual operation, and therefore only a lower capacity compressor of 1.74 kW is required.

Figure R1. Variation of hydrogen quality in the hydrogen storage tank

Figure R2. Variation of hydrogen import rate to the hydrogen storage tank

5. General - Figure's -the readability (Font size) needs improving.

Reply: Thank you very much for your advice. All of your suggestions are very important. To make it clearer for the reader, we enlarge the font size of the Figures in the new version, as shown in the example below.

(Line 192 to Line 193, Results)

Figure 2. The levelized cost of hydrogen and life cycle carbon emissions of multiple hydrogen production pathways in 31 provinces of China.

(Line 258 to Line 259, Results)

Figure 4. The optimal configuration of WE2, WE3, and WE4 in each province. (a) The optimal capacity of the WE2; (b) The optimal capacity of the WE3; (c) The optimal capacity of the WE4.

(Line 317 to Line 318, Results)

Figure 6. The impact of time-of-use electricity prices on the levelized cost of hydrogen for hydrogen production *via* water electrolysis. (a) WE1; (b) WE2; (c) WE3.

6. The manuscript will benefit from adding a conclusion.

Reply: Thanks for your helpful comments.

We supplement the conclusion to the end of the Introduction section in the new version, which is also in line with the format requirements of *Nature Communications*. “INTRODUCTION contain a brief summary of the major results and conclusions of the current work, written in the present tense”.

(Line 117 to Line 138, Introduction)

In this work, the analysis shows that the current LCOH of the four hydrogen production systems *via* water electrolysis in the provinces of China are 4.6 \$/kg H₂-7.9 \$/kg H₂, 4.6 \$/kg H₂-7.6 \$/kg H₂, 5.9 \$/kg H₂-10.1 \$/kg H₂, and 7.3 \$/kg H₂-14.8 \$/kg H₂, respectively. The LCOH is generally higher in the eastern and southern coastal areas, and is relatively lower in the western and northern regions. The implementation of time-of-use electricity prices can reduce the LCOH of the grid-connected hydrogen production systems *via* water electrolysis by 0.18 \$/kg H₂-0.90 \$/kg H₂. The LCCE of the four systems are 5.2 kg CO₂/kg H₂-59.3 kg CO₂/kg H₂, 4.3 kg CO₂/kg H₂-47.4 kg

CO₂/kg H₂, 3.7 kg CO₂/kg H₂-30.8 kg CO₂/kg H₂, and 2.22-2.28 kg CO₂/kg H₂, respectively. Grid-connected hydrogen production systems *via* water electrolysis have higher LCCE in the north, and have low carbon advantages only in some provinces, such as Qinghai, Tibet, Sichuan, and Yunnan. A total of 46.1 billion dollars of investment is required for China to achieve a hydrogen production structure transformation to 15% renewable hydrogen, which cumulatively reduces carbon emissions by 61.9 Mt. The LCOH of off-grid hydrogen production system *via* water electrolysis will be reduced to 2.2 \$/kg H₂ by 2045-2050, making it the most economical pathway to produce hydrogen, and certain carbon reduction incentives or production subsidy may enable this to happen 5-15 years earlier. In summary, the optimization results can provide guidance for investors in hydrogen production projects, the evaluation results of hydrogen production system in different provinces can provide reference for national and provincial energy policy makers.

7. Method section

a. Why is wind not included?

Reply: Thanks for your comment.

Mid-to-Long-Term Plan for the Development of Hydrogen Energy Industry (2021-2035) in China clearly proposed that the renewable energy hydrogen production to replace fossil energy hydrogen production in ammonia, methanol, refining and other industrial fields¹. Currently, hydrogen production equipment is mainly located within industrial plants, for example, about 80% of the hydrogen used in refineries was produced onsite at the refineries themselves^{2, 3}. Distributed hydrogen production is gaining more attention due to the fact that it is not plagued by long distance hydrogen transportation⁴. Beijing, Anhui and Guangdong, etc. have issued policies to support the distribution hydrogen production project construction explicitly to promote the hydrogen source supply nearby⁵. Compared to wind turbines, modular photovoltaic technology is more flexible, with power outputs ranging from W to MW, and is weakly constrained by geographic conditions and space^{6, 7}. Photovoltaics is the preferred option for the development of distributed renewable energy hydrogen production on the application side. For the above reasons, wind is not considered in this paper, and only distributed photovoltaic is considered. This point is clarified in the Methods section.

1. *Mid-to-Long-Term Plan for the Development of Hydrogen Energy Industry (2021-2035)*. https://zfxgk.nea.gov.cn/1310525630_16479984022991n.pdf (2022).
2. *Global Hydrogen Review 2023*. International Energy Agency (2023).
3. *Comparison of Commercial, State-of-the-Art, Fossil-Based Hydrogen Production Technologies*. National Energy Technology Laboratory. <https://www.osti.gov/servlets/purl/1862910/> (2022).
4. Handique AJ, Peer R, Haas J, Osorio-Aravena JC, Reyes-Chamorro L. *Distributed hydrogen systems: A literature review*. *International Journal of Hydrogen Energy* 85, 427-439 (2024).
5. *Several Policy Measures of Beijing Municipality on Supporting the Development*

of Hydrogen Energy Industry.
<https://www.beijing.gov.cn/zhengce/zhengcefagui/202208/W020220822333692689224.pdf> (2022).

6. Cheng G, et al. Analysis and prediction of green hydrogen production potential by photovoltaic-powered water electrolysis using machine learning in China. *Energy* **284**, (2023).
7. Achour Y, Berrada A, Arechkik A, El Mrabet R. Techno-Economic Assessment of hydrogen production from three different solar photovoltaic technologies. *International Journal of Hydrogen Energy* **48**, 32261-32276 (2023).

(Line 491 to Line 499, Methods)

System 2 (WE2): the hydrogen production system combined photovoltaic and grid. In this system, the electrical power of the electrolyzer comes from photovoltaics or the grid. The main motivation for this configuration is that this system can integrate distributed photovoltaics. Modular distributed photovoltaic technology makes hydrogen production systems easier to deploy near hydrogen applications^{30,31}. The grid is used as a backup power source, which ensures the hourly reliability of the hydrogen supply, thus eliminating the need to equip compression and energy storage³². It is worth noting that photovoltaic panels are not mandatory to install, depending on the results of optimization.

30. Cheng G, et al. Analysis and prediction of green hydrogen production potential by photovoltaic-powered water electrolysis using machine learning in China. *Energy* **284**, (2023).
31. Achour Y, Berrada A, Arechkik A, El Mrabet R. Techno-Economic Assessment of hydrogen production from three different solar photovoltaic technologies. *International Journal of Hydrogen Energy* **48**, 32261-32276 (2023).

b. Why is there a constraint of using hydrogen production rate at only 1kg/hr or 8760kg/year? Is this normalisation? Does the outcome change when economies of scale are considered?

Reply: Thank you very much for your advice. All of your suggestions are very important.

Firstly, it is important to clarify that in this paper, the hydrogen supply rate is limited to 1 kg/h, i.e., the reliability of the hydrogen supply is guaranteed. For WE1 and WE2, the supply rate is equal to the production rate due to the stable power supply. For WE3 and WE4, the power supply is not stable, so the hydrogen storage tanks are needed to regulate the stable supply of hydrogen, and the production rate of the electrolyzer is not fixed.

For the following reasons, constraining the hourly supply rate of hydrogen to a fixed value is necessary and normalized. First, each hydrogen production pathway needs to achieve the same hydrogen supply reliability (hydrogen supply rate) for a fair comparison of LCOH. Second, more than 90% of the global hydrogen used in industrial applications such as refining, ammonia synthesis, and methanol preparation, a stable

supply of hydrogen is an absolute requirement for safe industrial production. Most hydrogen production equipment is currently arranged in industrial plants with staggered downtime for maintenance to guarantee a continuous and stable supply of hydrogen.

In this paper, the fixed supply rate is assumed at 1kg/h with reference to other studies. A sensitivity analysis is conducted for hydrogen supply rates, which are set at 1 kg/h, 10 kg/h, 100 kg/h, 1000 kg/h, and 10000 kg/h, respectively. The results indicate that LCOH does not change with hydrogen supply rate. Therefore, this assumption is valid. This also suggests that considering changes in hydrogen production scale, the outcome does not change in this paper. Because the economic parameters used in this paper are all unit costs, for example, the cost unit of equipment such as photovoltaic panels and electrolyzer is \$/kW.

This is supplemented in the new version as follows.

(Line 472 to Line 485, Methods)

Overview of hydrogen production pathways

Each hydrogen production pathway needs to achieve the same hydrogen supply reliability (hydrogen supply rate) for a fair comparison of LCOH. More than 90% of the global hydrogen used in industrial applications such as refining, ammonia synthesis, and methanol preparation, a stable supply of hydrogen is an absolute requirement for safe industrial production¹². Most hydrogen production equipment is currently arranged in industrial plants with staggered downtime for maintenance to guarantee a continuous and stable supply of hydrogen^{1, 28}. For these reasons, it is necessary and important to limit the hourly hydrogen supply for each hydrogen production pathway to a fixed value²⁹. In this paper, it is assumed that the hydrogen supply rate is 1 kg/h. A sensitivity analysis is conducted for hydrogen supply rates, which are set at 1 kg/h, 10 kg/h, 100 kg/h, 1000 kg/h, and 10000 kg/h, respectively. The results are shown in Figure S7 of the Supplementary materials, which indicates that LCOH does not change with hydrogen supply rate. Therefore, this assumption is valid.

1. *Global Hydrogen Review 2023*. International Energy Agency (2023).
12. *Prospect of hydrogen energy industry in China*. Boston Consulting Group (2023).
28. *Comparison of Commercial, State-of-the-Art, Fossil-Based Hydrogen Production Technologies*. National Energy Technology Laboratory. <https://www.osti.gov/servlets/purl/1862910/> (2022).
29. Bracci JM, Sherwin ED, Boness NL, Brandt AR. A cost comparison of various hourly-reliable and net-zero hydrogen production pathways in the United States. *Nature Communications* 14, (2023).

(Supplementary materials)

S7. Sensitivity analysis of hydrogen supply rate

Figure S7. Sensitivity analysis of hydrogen supply rate. (The average LCOH in China as an example)

c. Lines 399-401. WE1. The capacity of the electrolyser is the same for each province so what is being optimized?

Reply: Sorry for the confusion. Producing 1 kg of hydrogen per hour requires 54.3 kW electrolyzer. For WE1, the capacity of the electrolyzer can be obtained directly by calculation. This is revised in the new version as follows.

(Line 487 to Line 490, Methods)

System 1 (WE1): the grid hydrogen production system. In this system, all the electric power of the electrolyzer comes from the grid, and the capacity of the electrolyzer can be obtained directly through calculation, *i.e.*, a 54.3 kW electrolyzer is required for the production per hour of 1 kg hydrogen.

d. Line 408. The statement “...there is no need for energy storage” needs explaining. Why are WE1 and WE2 configurations having no compression or storage? This implies an assumption that offtake use applications uses H₂ at the same rate of H₂ production and at the low hydrogen pressure provided from the electrolyser. Is this assumption valid in all provinces?

Reply: Thank you very much for your advice.

More than 90% of the global hydrogen used in industrial applications such as refining, ammonia synthesis, and methanol preparation, a stable supply of hydrogen is an absolute requirement for safe industrial production. Most hydrogen production equipment is currently arranged in industrial plants, and hydrogen is piped in at lower pressures. For example, about 80% of the hydrogen used in refineries was produced onsite at the refineries themselves. The hydrogen production equipment is staggered down for maintenance to guarantee a continuous and stable supply of hydrogen. For these reasons, it is assumed that end-user application of low-pressure and stable hydrogen is reasonable and necessary, and is valid in all provinces.

In addition, the grid is used as a backup power source in WE1 and WE2, which ensures the hourly reliability of the hydrogen supply, thus eliminating the need to equip compression and energy storage. Although WE3 is also connected to the grid, power purchases are constrained in order to ensure a 50% renewable energy percentage, so its power is fluctuating, thus it needs to be equipped with energy storage to ensure a stable supply of hydrogen.

These are supplemented in the new version as follows.

(Line 472 to Line 509, Methods)

Overview of hydrogen production pathways

Each hydrogen production pathway needs to achieve the same hydrogen supply reliability (hydrogen supply rate) for a fair comparison of LCOH. More than 90% of the global hydrogen used in industrial applications such as refining, ammonia synthesis, and methanol preparation, a stable supply of hydrogen is an absolute requirement for safe industrial production¹². Most hydrogen production equipment is currently arranged in industrial plants with staggered downtime for maintenance to guarantee a continuous and stable supply of hydrogen^{1, 28}. For these reasons, it is necessary and important to limit the hourly hydrogen supply for each hydrogen production pathway to a fixed value²⁹. In this paper, it is assumed that the hydrogen supply rate is 1 kg/h. A sensitivity analysis is conducted for hydrogen supply rates, which are set at 1 kg/h, 10 kg/h, 100 kg/h, 1000 kg/h, and 10000 kg/h, respectively. The results are shown in Figure S7 of the Supplementary materials, which indicates that LCOH does not change with hydrogen supply rate. Therefore, this assumption is valid.

Hydrogen production via water electrolysis

System 1 (WE1): the grid hydrogen production system. In this system, all the electric power of the electrolyzer comes from the grid, and the capacity of the electrolyzer can be obtained directly through calculation, *i.e.*, a 54.3 kW electrolyzer is required for the production per hour of 1 kg hydrogen.

System 2 (WE2): the hydrogen production system combined photovoltaic and grid. In this system, the electrical power of the electrolyzer comes from photovoltaics or the grid. The main motivation for this configuration is that this system can integrate distributed photovoltaics. Modular distributed photovoltaic technology makes hydrogen production systems easier to deploy near hydrogen applications^{30,31}. The grid is used as a backup power source, which ensures the hourly reliability of the hydrogen supply, thus eliminating the need to equip compression and energy storage³². It is worth noting that photovoltaic panels are not mandatory to install, depending on the results of optimization.

System 3 (WE3): the hydrogen production system with fixed renewable energy penetration. In this system, the renewable energy penetration is set at 50%. This constraint is added to reduce the carbon emissions generated from the hydrogen production *via* water electrolysis purchasing electricity from the grid. The electricity for the electrolyzer comes from the grid and photovoltaic panels, and the supply of electricity is regulated by the electricity energy storage, and the supply of hydrogen is

regulated by the hydrogen storage tank. A stable supply of hydrogen on an hourly basis can be ensured through the combination of these two energy storage methods ²⁹. It is worth noting that electricity energy storage and hydrogen storage tank are not necessarily installed in the system, this depends on the optimization results.

e. Line 392-393 “this study did not consider the actual hydrogen demand, which is different in each province.” However, is it true that the Western provinces are further from large population centres and may have a greater cost in storage and distribution to the eastern provinces that changes the LCOH and LCCE. The assumption in this work that off-take dynamics and quantity is the same in all provinces may need investigating in this manuscript rather than for future work.

Reply: Thank you very much for your advice. This suggestion has important guiding significance for our thesis writing and research work.

Firstly, we calculate the actual hydrogen demand in all provinces and investigate the investment and carbon emission reduction of hydrogen production structure transformation considering hydrogen demand, supplemented in the new version. Through the research, it is found that there is a mismatch in the spatial distribution of renewable hydrogen resources and hydrogen demand, with demand concentrated in the eastern provinces and renewable hydrogen resources concentrated in the western provinces.

Secondly, we strongly agree with you that the distribution costs of hydrogen may be higher in western provinces that are farther away from large population centers compared to eastern provinces. However, this does not change the LCOH and LCCE, as the focus of the study in this paper is on the comparison of LCOH and LCCE for the hydrogen production pathway excluding the distribution process.

Finally, combining the spatial distribution of renewable hydrogen resources and demand, we provide an outlook on solving the distribution mismatch through transmission and interaction of hydrogen resources between provinces.

(Line 319 to Line 352, Results)

Hydrogen production structural transformation considering demand

According to the National Energy Administration of China, the total demand for hydrogen in China is currently 33Mt. The total hydrogen demand is divided according to gross domestic product (GDP), population, and carbon emissions ²⁵. The hydrogen demand in each province is presented in Figure S6 in the Supplementary materials. Hydrogen demand in the provinces ranged from 0.05 Mt to 2.78 Mt, with significant differences. In the future, the share of hydrogen production *via* renewable energy sources will gradually increase, replacing some of the hydrogen production *via* fossil energy sources. Therefore, in this section, the investment and carbon reduction of transforming from the current to the future hydrogen production structure is quantified, considering the actual demand for hydrogen in each province. The future hydrogen production structure will have a 15% share of hydrogen produced *via* renewable energy sources, which is predicted by the China Hydrogen Alliance for the year 2030 ²⁶. It is

assumed that all provinces are required to complete the transition, and the WE3 and WE4 with lower carbon emissions are representative of hydrogen production *via* renewable energy sources.

A total of 46.1 billion \$ of investment is required for China to achieve a hydrogen production structure transformation to 15% renewable hydrogen, which cumulatively reduces carbon emissions by 61.9 Mt, with a carbon emission reduction rate of 10.1%. The investment and carbon reduction in each province is shown in Figure 7. It can be seen from the figure that provinces with high hydrogen demand require more investment for the transformation, while reducing more carbon emissions. The largest hydrogen demand is in Guangdong, Jiangsu, and Shandong, with 2.78 Mt, 2.59 Mt, and 2.53 Mt, respectively, and the investment required for the structural transformation of hydrogen production is 3.9 billion \$, 3.7 billion \$, and 3.1 billion \$, respectively, which reduces carbon emissions by 6.3 Mt, 4.5 Mt, and 4.2 Mt, respectively. It is worth noting that although the above provinces have the most emission reductions, the reduction rates are not the most advantageous, at 12.4%, 9.3%, and 8.8%, respectively. Higher carbon reduction rate can be achieved in provinces with abundant photovoltaic resources and low carbon emissions from the grid, such as Qinghai and Tibet, both at 17.0%, however, the hydrogen demand in these provinces is generally low, at 0.13 Mt and 0.05 Mt, respectively.

Figure 7. Investment and carbon emission reduction for hydrogen production structural transformation in provinces of China.

25. Pan G, *et al.* Feasibility of Scaling up the Cost-Competitive and Clean Electrolytic Hydrogen Supply in China. *Engineering* **39**, 154-165 (2024).
26. *China hydrogen energy and fuel cell industry white paper*. China Hydrogen Alliance.

(Supplementary materials)

S5. Distribution of hydrogen demand by province

Figure S6. Distribution of hydrogen demand by province

(Line 466 to Line 470, Discussion)

The mismatch between the spatial distribution of renewable hydrogen resources and demand can be observed through the distribution of carbon emission reduction and reduction rate in each province. How to solve the above problems of mismatch by utilizing low-cost and long-distance hydrogen transmission technologies is the focus of future study.

f. Line 410 WE3. "... renewable energy accounts for 50% of the total hydrogen production." Why is this constraint used rather than RE percentage from solar being a variable determined by optimisation?

Reply: Thank you very much for your review.

Firstly, the four systems WE1, WE2, WE3 and WE4 are setting up with the purpose of exploring the LCOH and LCCE of the hydrogen production system *via* water electrolysis under different renewable energy penetrations. The power of WE1 is completely from the grid and the renewable energy penetration is 0%. WE2 is based on the number of PV panels optimized to determine the renewable energy penetration. The power of WE4 is completely from renewable energy sources and the renewable energy penetration is 100%. WE3 is a compromise scenario, setting the renewable energy penetration to be 50%.

Secondly, the optimization objective of this paper is levelized cost of hydrogen production. If the renewable energy percentage is used as an optimization variable, its optimized result will be consistent with WE2, i.e., the grid is the most economical as a backup power source under the economic objective. Therefore, the constraint of 50% is chosen so that the hydrogen production system reduces power purchases from the grid and thus reduces carbon emissions.

This point is clarified in the new version.

(Line 500 to Line 503, Methods)

System 3 (WE3): the hydrogen production system with fixed renewable energy penetration. In this system, the renewable energy penetration is set at 50%. This constraint is added to reduce the carbon emissions generated from the hydrogen production *via* water electrolysis purchasing electricity from the grid.

(Line 518 to Line 521, Methods)

The four hydrogen production systems *via* water electrolysis have different renewable energy penetrations. The systems all use proton exchange membrane (PEM) electrolyzer to produce hydrogen by electrolyzing water, which has the advantages of green and flexible production, and high purity³⁵.

g. Line 438 “The reliability of hydrogen is ensured at a constant production rate per hour, which is necessary for hydrogen plants”. This implies an assumption that there is little/no degradation of electrolyser efficiency when operated at constant power. Is there also a case where the electrolyser technology can today or next generations have little/no degradation when power cycling and if so, what is the outcome when the electrolyser is operated dynamically? Or, what is the result when operating an electrolyser dynamically with a larger degradation rate compared to an electrolyser operating in constant power condition?

Reply: Thank you very much for your advice.

Firstly, sorry for the confusion. The reliability of hydrogen is ensured at a constant supply rate per hour, which is necessary for hydrogen using plants. This is because more than 90% of the global hydrogen used in industrial applications such as refining, ammonia synthesis, and methanol preparation, a stable supply of hydrogen is an absolute requirement for safe industrial production. Most hydrogen production equipment is currently arranged in industrial plants with staggered downtime for maintenance to guarantee a continuous and stable supply of hydrogen. The sentence is modified as follows.

(Line 530 to Line 531, Methods)

The reliability of hydrogen is ensured at a constant supply rate per hour, which is necessary for hydrogen using plants.

Secondly, based on the hydrogen supply reliability above, four hydrogen production systems *via* water electrolysis are set up. WE1 and WE2 have a stable power supply and can be operated at constant power at full load, while WE3 and WE4 do not have a stable power supply, so the electrolyzer are operated dynamically, and are equipped with the hydrogen storage tanks to ensure a stable supply of hydrogen.

Thirdly, there is degradation in both constant power operation or dynamic operation of the electrolyzer. The full-load operation hours and degradation rates of the electrolyzer for the four systems are supplemented in the new version. The degradation rates for WE3 and WE4 are 2.2% and 1.9%, respectively, with minimal impact on

lifetime.

Finally, the degradation rates of electrolyzer with the same capacity under constant and dynamic power operation are compared, taking WE4 as an example. And the degradation rates of the proton exchange membrane (PEM) and alkaline (ALK) electrolyzer are compared for the same operation conditions. The results show that the degradation rates of PEM electrolyzer are much smaller than that of ALK electrolyzer under the same operation condition, and dynamic power operation has less effect on PEM electrolyzer but increases the degradation rate of ALK electrolyzer. The rationale of adopting the proton exchange membrane (PEM) electrolyzer is validated.

These are supplemented in the new version.

(Line 247 to Line 258, Results)

In addition, compared to WE1 and WE2, the full load operation hours of the electrolyzer in WE3 and WE4 are substantially lower. In the case of Beijing, the annual full load operation hours in WE3 and WE4 are only 1679 h and 1107 h respectively, and the full load operation rates are 19.2% and 12.6%, the degradation rates are 2.2% and 1.9%, with only a small impact on the lifetime, as shown in Section S3 of the Supplementary materials. The degradation rates for constant and dynamic power operation at the same capacity are both 1.9%, using WE4 as an example. If the alkaline (ALK) electrolyzer is replaced, the degradation rates for the same operation condition will be 14.3% and 15.9%. The dynamic power operation has less effect on proton exchange membrane (PEM) electrolyzer but increases the degradation rate of ALK electrolyzer. And the rationale of adopting the PEM electrolyzer is validated.

(Supplementary materials)

S3. Operation characteristics of hydrogen production system *via* water electrolysis in Beijing

Figure S2. Hourly electric power of electrolyzer in a typical month.

(a) WE3; (b) WE4.

The degradation rate of the electrolyzer (D_{EL}) is calculated as follows ¹⁵.

$$D_{EL} = \frac{H_{sto,EL} + \frac{1}{h} \times \sum_{t=1}^{t=h} \left(\frac{EP_{EL}(t)}{IC_{EL}} \times (\sigma_{EL} - H_{sto,EL}) + k_{EL} f_{EL}(t)^2 \right)}{N_{EL}} \quad (S41)$$

$$f_{EL}(t) = \frac{|EP_{EL}(t) - EP_{EL}(t-1)|}{IC_{EL}} \quad (S42)$$

where, EP_{EL} is the electric power of the electrolyzer, kW. N_{EL} is the lifetime of the electrolyzer, year. $H_{sto,EL}$, σ_{EL} , k_{EL} are empirical coefficients related to the degradation rate, the proton exchange membrane (PEM) electrolyzer is taken as 0.35, 0.5, 1×10^{-5} , and the alkaline (ALK) electrolyzer is taken as 2, 10, 0.1, respectively. f_{EL} is the power volatility of the electrolyzer.

Table S8. Full load operation time and degradation rate of four hydrogen production systems *via* water electrolysis

System	WE1	WE2	WE3	WE4
Full load operation hours (h)	8760	8760	1679	1107
Full load operation rate (%)	100	100	19.2	12.6
Degradation rate (%)	2.5	2.5	2.2	1.9

Table S9. Comparison of electrolyzer degradation rates between constant power operation and dynamic operation

System	Constant power operation	Dynamic operation
PEM electrolyzer (%)	1.9	1.9
ALK electrolyzer (%)	14.3	15.9

(Taking WE4 as an example, the hydrogen production rate of the electrolyzer is 1kg/h in constant power operation.)

- Yu B, Fan G, Sun K, Chen J, Sun B, Tian P. Adaptive energy optimization strategy of island renewable power-to-hydrogen system with hybrid electrolyzers structure. *Energy* **301**, (2024).